# POST-TRAINING QUANTIZATION IS ALL YOU NEED TO PERFORM CROSS-PLATFORM LEARNED IMAGE COMPRESSION

## ABSTRACT

It has been witnessed that learned image compression has outperformed conventional image coding techniques and tends to be practical in industrial applications. One of the most critical issues that need to be considered is the non-deterministic calculation, which makes the probability prediction cross-platform inconsistent and frustrates successful decoding. We propose to solve this problem by introducing well-developed post-training quantization and making the model inference integer-arithmetic-only, which is much simpler than presently existing training and fine-tuning based approaches yet still keeps the superior rate-distortion performance of learned image compression. Based on that, we further improve the discretization of the entropy parameters and extend the deterministic inference to fit Gaussian mixture models. With our proposed methods, the current state-of-the-art image compression models can infer in a cross-platform consistent manner, which makes the further development and practice of learned image compression more promising.

## 1 INTRODUCTION

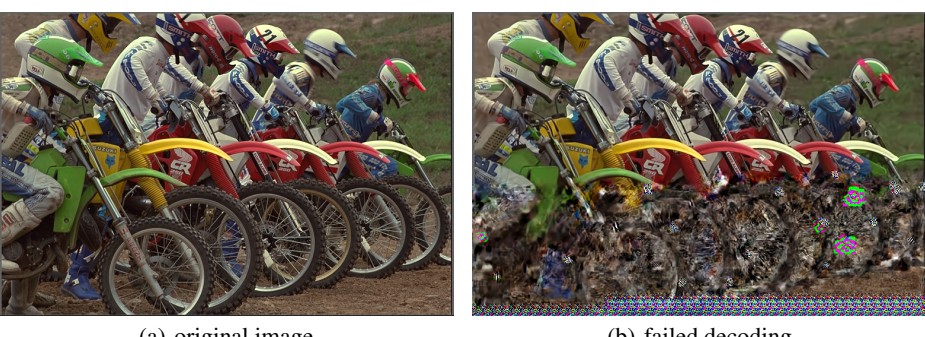

(a) original image          (b) failed decoding

Figure 1: The cross-platform inconsistency caused by non-deterministic model inference. This inconsistency is a catastrophe to establish general-purpose image compression systems, and it is almost inevitable when practicing learned image compression with floating-point arithmetic.

In recent years, learned data compression techniques have attracted a lot of attention and achieved remarkable progress (Ballé et al., 2017; 2018; Minnen et al., 2018; Cheng et al., 2020; He et al., 2021; Guo et al., 2021). Recently, a few learned lossy image compression approaches (Minnen et al., 2018; Cheng et al., 2020; He et al., 2021; Guo et al., 2021) have outperformed BPG (Bellard, 2015) and the intra-frame coding of VVC (vtm, 2020), which are the state-of-the-art manually designed image compression algorithms. As lossy image compression is one of the most fundamental techniques of visual data encoding, these results promise possibility of transmitting and storing images or frames at a lower bit rate, which can benefit almost all industrial applications dealing with visual data. Therefore, it is highly important to study how to make these rapidly developing learning-based methods practical.

Most currently state-of-the-art learned image compression approaches adopt a forward-adaptive coding scheme proposed in Ballé et al. (2018). The model will firstly transform the image to a major representation $\hat{y}$ and a minor representation $\hat{z}$. Then $\hat{y}$ will be encoded by an arithmetic coder (Rissanen & Langdon, 1981; Martin, 1979) with independently saved $\hat{z}$ as side-information. Later this method is further delved by introducing backward adaption with context modeling (Minnen et al., 2018; He et al., 2021; Guo et al., 2021). However, this line of models suffer from a non-determinism issue (Ballé et al., 2019), which makes the image hard to be decoded when the encoder and the decoder run on heterogeneous platforms. The non-determinism is caused by floating-point arithmetic, which can let the calculation results, from the same input operands but on different software or hardware platforms, different. This issue makes calculating prior from side-information and context model cross-platform inconsistent so that the decoding fails (See Figure 1). As cross-platform transmission is a vital requirement of establishing practical image coding systems, this inconsistency is critical.

To make the computation deterministic for cross-platform consistency, Ballé et al. (2019) proposes to train an integer network specially designed for learned image compression, where the inference stage is integer-arithmetic-only. It performs well on earlier proposed models like Ballé et al. (2018). After determinizing Ballé et al. (2018) with integer network, the compression performance has a marginal reduction compared with the floating-point version. However, we find that on more complex models with context modeling like Minnen et al. (2018) and Cheng et al. (2020), adopting this integer network approach cannot keep the performance loss negligible. Another previous work (Sun et al. (2021)) proposes to quantize model parameters and activation to fix-point, enabling cross-platform decoding of the mean-scale hyperprior-only network in Minnen et al. (2018) without hurting compression performance. However, the determinism of state-of-the-art joint autoregressive and hyperprior methods (Minnen et al., 2018; Cheng et al., 2020; He et al., 2021; Guo et al., 2021) are still not considered. As the slow serial decoding problem of autoregressive context model has been addressed in He et al. (2021), this joint modeling architecture is very promising and valuable for practical application and its cross-platform decoding issue needs to be solved.

We notice that, these existing approaches are similar to general model quantization techniques in spirit, *i.e.* quantization-aware training (QAT, Jacob et al. (2018); Krishnamoorthi (2018); Esser et al. (2019); Bhalgat et al. (2020)) and post-training quantization (PTQ, Nagel et al. (2019; 2020); Li et al. (2020)). Instead of proposing another approach to provide an ad-hoc solution, we advocate to solve this cross-platform consistency issue based on those well-developed model quantization techniques in a more flexible and extendable manner. Thus, we investigate cross-platform consistent inference with PTQ. Usually, it requires much less calibration data (Nagel et al., 2020; 2021; Li et al., 2020) and costs much less time to quantize the model than QAT, yet for several computer vision tasks it can achieve almost the same accuracy as QAT when the target bit-width is 8 (Nagel et al., 2019; 2020; Li et al., 2020). It does not demand re-training or fine-tuning the trained floating-point models, which is very friendly to industrial deployment. By applying integer-arithmetic-only operators (Jacob et al., 2018; Zhao et al., 2020; Yao et al., 2021) after quantization, we can achieve deterministic models which get rid of the inconsistency issue.

In this paper, we contribute to the community from following perspectives:

1. We evaluate and prove that, the deterministic computing issue of learned data compression can be reduced to a general model quantization problem. After applying a standard post-training quantization (PTQ) technique, we obtain cross-platform consistent image compression models with marginal compression performance loss.

2. We successfully determinize presently state-of-the-art learned image compression models with context modeling and Gaussian mixture models. To the best of our knowledge, this is the first work investigating and accessing cross-platform consistent inference on those models. As we use standard quantization techniques, it is also promising to adopt our method to determinize future compression models.

3. We propose a novel approach to discretize the entropy parameters, which can be computed directly in a deterministic manner. Compared with the existing method based on searching algorithm, our method significantly speeds up the parameter discretization and eliminates the bottleneck.

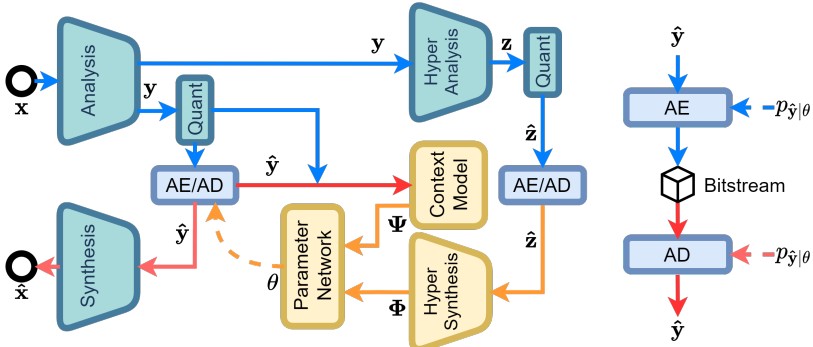

Figure 2: Diagram of joint autoregressive and hyperprior architecture for learned image compression (Minnen et al., 2018; Cheng et al., 2020). AE/AD denote arithmetic en/de-coder. $\theta$ is the set of predicted entropy parameters (*i.e.* $\pi, \mu, \sigma$), modeling the distribution of $\hat{\mathbf{y}}$ element-wisely. The blue, red and orange arrows denote encoding, decoding and shared data flows, respectively. The highlighted orange networks are shared by both encoding and decoding to estimate code probability for AE/AD, demanding cross-platform consistency.

We emphasize that our goal is not to propose a new method for quantizing neural networks, but to call attention to the relation between the deterministic issue and model quantization techniques. Benefiting from the maturely developed quantization techniques, we show that the cross-platform consistency issue in learned image compression can be better solved.

## 2 PRELIMINARY

**Notations.**

We represent matrices with capital bold letters (*e.g.* $\mathbf{W}$) and denote vectors as small bold letters (*e.g.* $\mathbf{v}$). We use ceiling $\lceil \cdot \rceil$ and floor $\lfloor \cdot \rfloor$ notations to represent round-up and round-down operators, respectively. And the blended $\lceil \cdot \rfloor$ denotes round-to-nearest. By default, we denote the inputs and activated outputs of a linear layer as $\mathbf{v}$ and $\mathbf{u}$ respectively. And the activation function is represented by $h(\cdot)$. We use the capital $B$ as the quantization bit-width, which by default is set to 8 as we prefer an 8-bit quantization. We use scalar $s$ with a subscript to denote the corresponding quantization step, *e.g.* $s_{\mathbf{v}}$ means the quantization step of $\mathbf{v}$. We may introduce a bracketed superscript to distinguish variables in different layers, *e.g.* $\mathbf{v}^{(\ell)}$ is the input of the $\ell$-th layer while $\mathbf{v}^{(\ell+1)}$ is the input of the next layer. For simplicity, we use the notation $\text{clip}(\cdot)$ as value clipping introduced by uniform affine quantization, omitting the clipping bounds. By default we quantize the tensors to $B$-bit signed integers, with the upper and lower clipping bounds $2^{B-1} - 1$ and $-2^{B-1}$.

**Learned image compression and non-determinism issue.**

In Appendix D, we describe several popular deep-learning-based image compression methods. Briefly speaking, almost all the cutting-edge learned image compression techniques adopt the joint autoregressive and hyperprior modeling illustrated in Figure 2. To encode the major symbols $\hat{y}_i$, an entropy model $p_{\hat{\mathbf{y}}|\hat{\mathbf{z}}, \hat{\mathbf{y}}_{j<i}}(\hat{y}_i; \hat{\mathbf{z}}, \hat{\mathbf{y}}_{j<i})$ is adopted to predict the probability density of $\hat{y}_i$ conditioned on side-information $\hat{\mathbf{z}}$ and already decoded symbols $\hat{\mathbf{y}}_{j<i}$. The predicted density functions will then be accumulated to obtain the cumulative distribution functions (CDF) and fed to arithmetic en/de-coders.

Generally, we input $\hat{\mathbf{z}}$ to the *hyper synthesizer* to obtain the intermediate feature $\mathbf{\Phi}$, and use the *context models* to summarize the context representation $\mathbf{\Psi}$. Then, we use a *parameter network* to calculate the element-wise distribution parameters from $\mathbf{\Phi}$ and $\mathbf{\Psi}$ to generate the predicted distributions. To ensure correct decompression, we should make sure the inference of above-mentioned models (*hyper synthesizer*, *context models* and *parameter network*) *deterministic*. It means that, with the same input $\hat{\mathbf{z}}$ and $\hat{\mathbf{y}}_{j<i}$, the predicted entropy parameters $\theta$ should always be the same no matter what platform the encoder or decoder is running on. The calculation should be platform-independent

and cross-platform consistent. However, if the models infer with floating-point numbers, the determinism is hard to satisfy.

**Uniform affine quantization (UAQ).** UAQ is widely adopted by model quantization researches (Jacob et al., 2018; Nagel et al., 2019; 2020; Li et al., 2020). In UAQ, both activation values and weights should be quantized to fixed point numbers to allow the use of integer matrix multiplications. Usually, UAQ maps floating-point values to $B$-bit integers. For a given vector $\mathbf{v}$, UAQ with a quantization step $s_{\mathbf{v}}$ is formulated as:

$$\mathbf{q_v} = \text{clip}\left(\left\lceil s_{\mathbf{v}}^{-1}\mathbf{v}\right\rfloor + z_{\mathbf{v}}\right) \tag{1}$$

The term $z_{\mathbf{v}}$ is an integer representing the zero-point shifting the center of quantization range, which is applied in asymmetric uniform affine quantization and omitted in symmetric ones. In recent QAT, the step $s_{\mathbf{v}}$ and zero-point $z_{\mathbf{v}}$ are usually learned during training/fine-tuning (Esser et al., 2019). And in PTQ they are normally determined by the value range of given weights or activation vectors (Nagel et al., 2019; 2021).

Therefore, the floating point vector $\mathbf{v}$ is discretized to an integer vector $\mathbf{q_v}$, where each integer value actually represents a fixed point value. We can remap $\mathbf{q_v}$ to corresponding fix-point vector through dequantization:

$$\hat{\mathbf{v}} = s_{\mathbf{v}}\mathbf{q_v} - s_{\mathbf{v}}z_{\mathbf{v}} \tag{2}$$

When $\mathbf{v}$ is the activation output, the second term can be pre-computed and absorbed by the convolution bias, so it will not introduce extra calculation to inference. Therefore, it is recommended by Nagel et al. (2021) to apply asymmetric quantization on activation and symmetric quantization on weights. Nagel et al. (2021) also suggests to adopt a per-channel quantization on weights (Krishnamoorthi, 2018; Li et al., 2019), which we introduce in Appendix A.1.

**Integer-arithmetic-only requantization.** Several previous works on QAT for integer-arithmetic-only inference have been proposed (Jacob et al., 2018; Zhao et al., 2020; Yao et al., 2021). One of the most important techniques is the dyadic requantization. Between two convolution layers, the dequantized fixed point activation will get quantized again, this can be fused to one operation called requantization. A requantization after the $\ell$-th layer is:

$$\mathbf{q_v}^{(\ell+1)} = \text{clip}\left(\left\lceil m^{(\ell)}\mathbf{q_u}^{(\ell)}\right\rfloor + z_{\mathbf{u}}^{(\ell)}\right) \tag{3}$$

where $\mathbf{q_u}^{(\ell)}$ is the int32 activation accumulation of the $\ell$-th layer, $z_{\mathbf{u}}^{(\ell)}$ is the zero point, and $m^{(\ell)}$ is the requantization scale factor. Please refer to Appendix A.2 for detailed derivation of this equation.

Ideally, all floating-point operator should be removed from model inference but $m$ is still floating-point, which makes the requantization non-deterministic. Therefore, a dyadic number $m_0$ is introduced in Jacob et al. (2018) to approximate $m$:

$$m_0 = \left\lceil 2^n m\right\rfloor \tag{4}$$

where $m_0$ and $n$ are integers. Now we have $m \approx 2^{-n}m_0$. Obviously, the approximation error becomes smaller when $n$ gets larger. With this approximation, the requantization described in eq. 3 can be rewritten as:

$$\mathbf{q_v}^{(\ell+1)} = \text{clip}\left(\left\lceil \frac{m_0^{(\ell)}\mathbf{q_u}^{(\ell)}}{2^{n^{(\ell)}}}\right\rfloor + z_{\mathbf{u}}^{(\ell)}\right) = \text{clip}\left(\left\lceil \left(m_0^{(\ell)}\mathbf{q_u}^{(\ell)}\right)//2^{n^{(\ell)}}\right\rfloor + z_{\mathbf{u}}^{(\ell)}\right) \tag{5}$$

where the operator $//$ is integer division with rounding-to-nearest (round-int-div, RID). And this RID calculation can be further implemented as bit shift operation with rounding-to-nearest (round-int-shift, RIS). Since both $m_0$ and 32-bit activation $\mathbf{q_u}^{\ell}$ are integers now, the calculation of $\mathbf{q_v}^{(\ell+1)}$ requires only an integer multiplication, a RID or RIS and a clipping. Hence the requantization becomes integer-arithmetic-only.

## 3 Integer-only PTQ for deterministic coding

Different from works on general model quantization whose major purpose is to compress the model size and improve inference efficiency, we introduce quantization to eliminate the inconsistent arithmetic. Thus, we should strictly prevent using platform-specific operators. So we restrict all inference calculations to 32-bit integer arithmetic, as it is the most widely supported set of integer arithmetic by hardware devices. Particularly, leveraging execution efficiency and compatibility, we only consider adopting matrix multiplication on 8-bit integer operands to perform convolution.

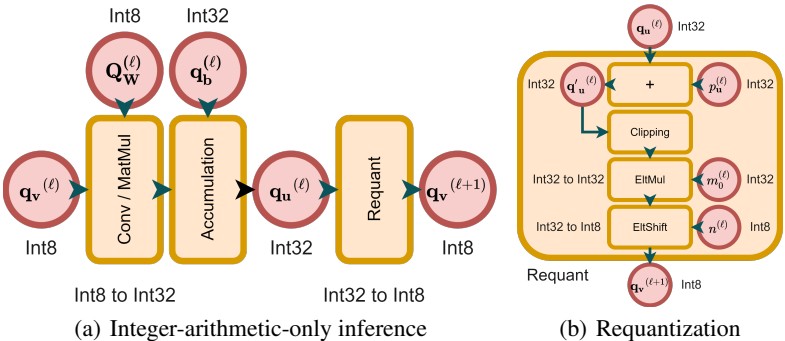

(a) Integer-arithmetic-only inference   (b) Requantization

Figure 3: The integer-arithmetic-only inference. An offline-constrained integer-arithmetic-only re-quantization will be adopted during inference, which we will discuss in Section 3.2.

### 3.1 PTQ BASELINE

We adopt a relatively simple and standard quantization pipeline as described in Nagel et al. (2021). We apply symmetric per-channel quantization on weights and asymmetric per-tensor quantization on activation except the output of the last layer in parameter net. Instead, we symmetrically quantize the activation from the last layer to 16 bits, with a fixed quantization step $2^{-6}$. This benefits the CDF indexing and calculation which we will discuss later in Section 4.1. As recommended in Nagel et al. (2021), we adopt a grid search minimizing the reconstruction error to obtain the weight quantization step $s_{\mathbf{W}}$ of each layer. We figure out the activation quantization step $s_{\mathbf{u}}$ with Min-Max method on a little calibration data. We detailedly describe the algorithms in Appendix C.3.

After setting the quantizers, we apply a per-block (or per-network) adaptive rounding reconstruction (Brecq, Li et al. (2020)), though the quantized models without this reconstruction have already achieved acceptable rate-distortion performance (Appendix C.4).

### 3.2 OFFLINE-CONSTRAINED INTEGER-ARITHMETIC-ONLY REQUANTIZATION

Previous use of integer-arithmetic-only requantization is based on QAT (Jacob et al., 2018; Yao et al., 2021; Zhao et al., 2020). It is first proposed in Jacob et al. (2018) where $m_0$ in eq. 4 is always set to an integer bigger than $2^{30}$ which should be represented with 32 bits. We find it risky, as this 32-bit integer $m_0$ will later multiply with the 32-bit integer accumulation. Assuming that no specific hardware support is available, we tend to believe that the requantization should be implemented with standard 32-bit integer arithmetic (*i.e.* we cannot conduct the bit-shift on 64-bit multiplication result registers). Thus, the 32-bit integer multiplication may overflow and result in error if we use unconstrained $m_0$. Another issue is, in Jacob et al. (2018) the requantization scale factor $m$ is empirically assumed as always less than 1, which is hard to satisfy in PTQ.

To address this problem, we slightly modify the requantization by moving the zero-point adding operation in eq. 3 forward:

$$p_{\mathbf{u}}^{(\ell)} = \left\lceil \frac{z_{\mathbf{u}}^{(\ell)}}{m^{(\ell)}} \right\rceil \tag{6}$$

$$\mathbf{q}_{\mathbf{v}}^{(\ell+1)} = \text{clip}\left(\left\lceil m^{(\ell)}\left(\mathbf{q}_{\mathbf{u}}^{(\ell)} + p_{\mathbf{u}}^{(\ell)}\right)\right\rceil\right) = \text{clip}\left(\left\lceil m^{(\ell)} \mathbf{q'}_{\mathbf{u}}^{(\ell)} \right\rceil\right) \tag{7}$$

where $\mathbf{q'}_{\mathbf{u}}^{(\ell)} = \mathbf{q}_{\mathbf{u}}^{(\ell)} + p_{\mathbf{u}}^{(\ell)}$ is the 32-bit integer activation biased with the 32-bit pre-scaling zero-point $p_{\mathbf{u}}^{(\ell)}$. By doing so, the asymmetric quantization of $\mathbf{q}_{\mathbf{u}}$ changes to the symmetric quantization of $\mathbf{q'}_{\mathbf{u}}$.

As the requantized value will be clipped to $B$-bit integer within a range $[-2^{B-1}, 2^{B-1} - 1]$, the maximal and minimal valid value of $\mathbf{q'}_{\mathbf{u}}^{(\ell)}$ are computable:

$$q_{\max}^{(\ell)} = \left\lfloor \frac{2^{B-1} - 1}{m^{(\ell)}} \right\rfloor, \quad q_{\min}^{(\ell)} = \left\lceil \frac{-2^{B-1}}{m^{(\ell)}} \right\rceil \tag{8}$$

with these bounds, we can earlier conduct the clipping operation before the rescaling (see Figure 6 in Appendix B.1), restricting the biased activation values $\mathbf{q'_u}$ in the range $[q_{\min}, q_{\max}]$. Thus, we can figure out the largest $n$ subject to $\forall q'_{\mathbf{u}} \in \mathbf{q'_u}, -2^{31} \leq m_0 q'_{\mathbf{u}} < 2^{31}$, which avoids overflow while keeps precision as much as possible:

$$n^{(\ell)} = 32 - B, \quad m_0^{(\ell)} = \left\lfloor 2^{n^{(\ell)}} m^{(\ell)} \right\rfloor \tag{9}$$

Please refer to Appendix B.1 for a detailed explanation of eq. 9. Therefore, after setting the quantizers, we offline calculate the dyadic numbers and replace the requantization operations layer by layer. Empirically, the proposed integer-arithmetic-only requantization with constrained magnitude of $m_0$ still keeps the model performance, though it introduces slightly more numerical error than the original version proposed in Jacob et al. (2018). We can safely adopt it across platforms with standard 32-bit integer multiplications and RISs/RIDs. Figure 3 (right) shows this offline-constrained form of requantization.

# 4 FROM 16-BIT OUTPUT TO DISCRETIZED CUMULATIVE DISTRIBUTION

## 4.1 BINARY LOGARITHM STD DISCRETIZATION FOR DETERMINISTIC ENTROPY MODELING

The network-output parameters should be discretized, so that the CDFs can be stored as limited look-up-tables (LUTs), as discussed in Appendix D.2. In previously proposed approaches (Ballé et al., 2019; Sun et al., 2021), the standard deviations (STD) are logarithmically discretized with computing natural logarithm $\log(\cdot)$:

$$\hat{i}_\sigma = \left\lfloor \frac{\log(\sigma) - \log(\sigma_{\min})}{\Delta_\sigma} \right\rfloor \tag{10}$$

which is hard to determinize. Sun et al. (2021) obtains $\hat{i}_\sigma$ from parameter network output in a deterministic way by comparing the fix-point $\sigma$ value with pre-computed sampling points $\hat{\sigma}$. This is moderately efficient as there are 64 levels of $\hat{\sigma}$ for each predicted STD to compare. We detailedly introduce this discretization approach in Appendix D.3 (eq. 51 and the following paragraph).

Empirically, the STD is long-tail distributed as most of the latents are predicted to have entropy close to zero. So the logarithmic discretization of $\sigma$ is reasonable to suppress error and has been proved effective. To keep this favor and be more hardware-friendly, we propose to use binary logarithm. Hence, we modify eq. 10 (and eq. 51 in Appendix D.3) to:

$$\Delta = \frac{1}{L-1} \log_2 \left( \frac{\sigma_{\max}}{\sigma_{\min}} \right)$$
$$\hat{i}_\sigma = \left\lfloor (\log_2(\sigma) - \log_2(\sigma_{\min})) \Delta^{-1} \right\rfloor \tag{11}$$
$$\hat{\sigma} = \sigma_{\min} \left( \exp2\left(\hat{i}\right) \right)^\Delta$$

where $\exp2(x) = 2^x$ denotes the power of 2. Ballé et al. (2019) and Sun et al. (2021) adopt the same bounds $\sigma_{\min} = 0.11$ and $\sigma_{\max} = 256$ with level $L = 64$. To omit non-determinism while to simplify the calculation, we instead adopt $\sigma_{\min} = 0.125, \sigma_{\max} = 32$ and $L = 9$. Therefore, we obtain a simplified formula with $\Delta = 1$ and $\log_2(\sigma_{\min}) = -3$. Since the STD has an extremely long-tail distribution, using a smaller upper bound $\sigma_{\max}$ is painless. Adopting less levels, however, enlarges error because the sampling points get sparser. Thus, we uniformly interpolate additional 7 minor levels between each two adjacent major levels of $\hat{\sigma}$. Considering the input $\sigma$ is the dequantized fix-point value with corresponding quantized integer $q$ and scale $s = 2^{-6}$ (i.e. $\sigma = sq$), the updated formula is:

$$\hat{i} = \lfloor \log_2(q) \rfloor + \log_2(s) + 3$$
$$\hat{j} = \left\lceil \frac{q - \exp2\left(\lfloor \log_2(q) \rfloor\right)}{\exp2\left(\lfloor \log_2(q) \rfloor - 3\right)} \right\rceil$$
$$\hat{i}_\sigma = 8\hat{i} + \hat{j} \tag{12}$$
$$\hat{\sigma} = \sigma_{\min} \left( \exp2\left(\hat{i}\right) + \hat{j}\exp2\left(\hat{i} - 3\right) \right)$$

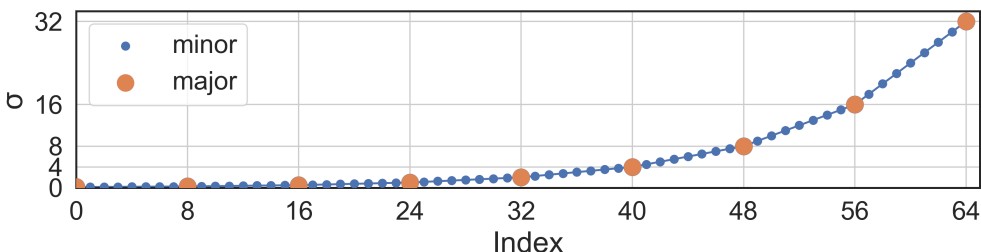

Figure 4: Visualization of the proposed 65-level STD parameter discretization. The orange dots denote binary-logarithmically distributed major levels and the blue ones are linearly interpolated minor levels.

where $\log_2(s) = -6$ is a constant integer. $\hat{i} \in \{0, 1, \ldots, 8\}$ and $\hat{j} \in \{0, 1, \ldots, 8\}$ are the major and minor indexes, respectively. Note that $\hat{i} = 8$ iff. $q = 2^{11}$, when $\hat{j} = 0$. Thus, the summary index $\hat{i}_\sigma \in \{0, 1, \ldots, 64\}$ corresponds to 65 values of $\hat{\sigma}$, which generates 65 CDFs to be stored as LUTs. We further describe the derivation of this discretization in Appendix B.2. Figure 4 shows the value distribution of this discretization.

Thus, the calculation of $\hat{i}_\sigma$ reduces to calculating the round-down integer binary logarithm of 16-bit integer $q$. To compute the binary-logarithm of an integer, we can adopt platform-specific instructions like `BSR` on x86 or `CLZ` on ARM. We can also use a platform-independent bit-wise algorithm to count the leading zeros to obtain result of integer binary logarithm, which is easy to vectorize. The Algorithm 1 in Appendix E.1 describes the detailed process converting the 16-bit output to the indexes.

## 4.2 Deterministic entropy coding with Gaussian mixture model

For models using GMM with $k$ Gaussian components, we can calculate CDF of the $i$-th component with given $\mu_i$ and $\sigma_i$ and accumulate it with a multiplier $\pi_i$. The predicted multiplier is dequantized 16-bit fix-point number $\pi_i = s_\pi q_{\pi_i}$:

$$C_{\text{GMM}(\hat{y})}(\hat{y}; \pi, \mu, \sigma) = \sum_{i=1}^{k} s_\pi q_{\pi_i} C_{\hat{y}}(\hat{y}; \mu_i, \sigma_i) \tag{13}$$

where $s_\pi$ is the quantization scale of $q_{\pi_i}$. We can omit $s_\pi$ in eq. 13 because it is a constant scalar normalizer and can be merged into the normalization factor of frequency based CDF table. The remaining part only involves integer arithmetic.

To obtain the CDF of the $i$-th Gaussian component $C_{\hat{y}}(\hat{y}; \mu_i, \sigma_i)$, we can query two-level pre-computed LUTs with combined $\hat{i}_\sigma$ (eq. 12) and $\hat{i}_\mu$ (eq. 53 in Appendix D.3) as outer index and $\hat{y} - \lfloor \mu_i \rfloor$ in eq. 52 as inner index. However, indexing the inner level LUT with $\hat{y} - \lfloor \mu_i \rfloor$ suffers from a risk of out-of-bound error. We cannot directly restrict the range of $\hat{y}$ to $[\lfloor \mu_i \rfloor - R, \lfloor \mu_i \rfloor + R]$ like Sun et al. (2021) because multiple Gaussian components are involved. If $\exists \mu_j$ subject to $\mu_i - \mu_j > R$, the direct restriction may result in significant error. Instead, we set the cumulative distribution value to 0 and $\text{CDF}_{\max}$ when $\hat{y} - \lfloor \mu_i \rfloor \leq -R$ and $\hat{y} - \lfloor \mu_i \rfloor \geq R$ respectively. We restrict the symbol $\hat{y}$ no bigger than $(\max \lfloor \mu \rfloor) + R$ where the CDF value is $\text{CDF}_{\max}$. Also we restrict it larger than $(\min \lfloor \mu \rfloor) - R$ where the CDF value is zero. In situations the symbol $\hat{y}$ is out of these upper and lower bounds, we will adopt Golomb coding to encode $\hat{y}$, inspired by tensorflow-compression implementation[1]. This situation is quite rare as its corresponding probability mass is close to zero, and adopting Golomb coding will not damage the overall bit rate. Thus, the CDF of GMM in eq. 13 is still monotonically increasing which can be reversed with searching algorithms during decoding.

With pre-computed CDFs of single mu-scale Gaussian entropy model, probability of $\hat{y}_i$ in each Gaussian component can be checked from shared LUTs. Then the aggregated CDF value of $\hat{y}_i$ can

---

[1] `https://github.com/tensorflow/compression/blob/v2.2/tensorflow_compression/python/entropy_models/continuous_base.py#L80-L81`

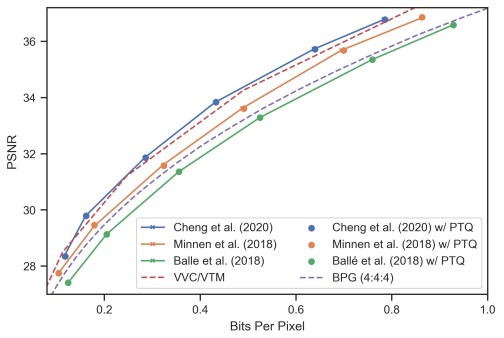 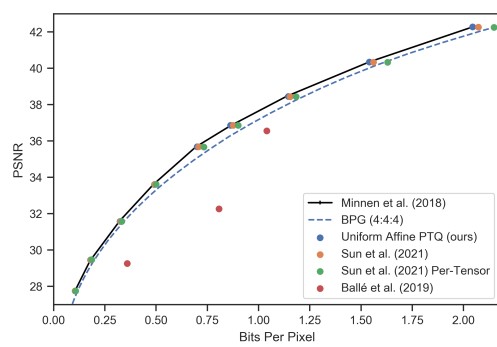

(a) Quantized models versus full-precision models.     (b) Comparison with prior works.

Figure 5: RD curves evaluated on Kodak. (a) RD performance of different models w/ or w/o quantization. The solid lines with cross markers denote original model inference with floating point numbers. The dots represent quantized 8-bit integer-arithmetic-only models, sharing the same color as its floating-point version. (b) We test Sun et al. (2021) by quantizing the activations in both per-channel and per-tensor manner. We also test Ballé et al. (2019).

be calculated with given $q_\pi$. In Appendix E.2, we provide pseudo-code of this GMM involving coding for reference. Algorithm 3 describes the whole encoding process and Algorithm 4 describes the decoding.

## 5 EXPERIMENTS

We train various presently state-of-the-art learned image compression architectures on floating point to obtain the full-precision models. After the training, we use PTQ algorithms to quantize sub-networks involved in the entropy estimation to 8-bit, as above-mentioned. Then we replace all requantization and Leaky ReLU (Appendix A.3) with corresponding deterministic adaptions and insert additional layers for LUT-index calculation. The detailed experiment setting is described in Appendix C.

### 5.1 COMPRESSION PERFORMANCE

| Method | ours | Sun et al. (2021) | |
|---|---|---|---|
| Act. Quant. | per-tensor | per-channel | per-tensor |
| BD-Rate (%) | **0.35** | 1.22 | 4.04 |

Table 1: BD-rates over full-precision model. The RD data is the same as that in Figure 5(b).

We quantize sub-networks of Ballé et al. (2018), Minnen et al. (2018), and Cheng et al. (2020) involving entropy prediction to 8-bit and compare their rate-distortion (RD) performance with original full-precision version, shown in Figure 5(a). The results indicate that, existing learned image compression techniques are compatible with standard PTQ. All tested models can infer with integer-arithmetic-only computations with negligible reduction on RD performance, which guarantees painless cross-platform consistency. It is not necessary any more for us to particularly develop new training techniques nor network structures to address the inconsistency issue.

We compare our approach with existing ones (Sun et al., 2021; Ballé et al., 2019). We present the RD results in Figure 5(b), and further report the corresponding BD-rates (Bjontegaard (2001)) in Table 1. When adopting our approach or Sun et al. (2021) on Minnen et al. (2018) model, the RD performance marginally deteriorates. However, Sun et al. (2021) adopts a per-channel activation quantization which is unfriendly to hardware implementation and rarely supported (Nagel et al., 2021). When adopting Sun et al. (2021) with per-tensor activation quantization, the performance, especially at higher bit-rates, gets hurt significantly. We have successfully reproduced the good

performance reported in Ballé et al. (2019) for determinizing Ballé et al. (2018) (not shown here), although we try hard, we find this method cannot keep marginal performance deterioration for determinizing context-model-involved architectures like Minnen et al. (2018). Note that Ballé et al. (2019) is a dedicated QAT method, we find it hard to train when applied on Minnen et al. (2018).

## 5.2 RATE OF DECODING ERROR

| PTQ | w/o PTQ (FP32) | | w/ PTQ (Int8) | |
|---|---|---|---|---|
| Encoding Platform | GPU | CPU | GPU | CPU |
| Error Rate on Kodak | 12/24 (50.0%) | 12/24 (50.0%) | **0/24 (0.0%)** | **0/24 (0.0%)** |
| Error Rate on Tecnick | 72/100 (72.0%) | 84/100 (84.0%) | **0/100 (0.0%)** | **0/100 (0.0%)** |

Table 2: Decoding error rates of Minnen et al. (2018) when inference with or without PTQ. The results are tested on NVIDIA GTX 1060 (marked as *GPU*) and Intel Core i7-7700 (marked as *CPU*). The decoding is cross-evaluated on the two platforms, *i.e.* encoding on one and decoding on the other.

Following Ballé et al. (2019) and Sun et al. (2021), we report the rate of decoding error in Table 2 for completeness. As the networks have been strictly restricted to only perform 8-bit and 32-bit integer arithmetic, the inference is strictly deterministic.

## 5.3 LATENCY OF BINARY LOGARITHM BASED STD DISCRETIZATION

| Method | Comparison (CompressAI) | Comparison (vectorized) | Calculation (ours) | Hyper Synthesis |
|---|---|---|---|---|
| Latency on Kodak | 17.32 | 9.52 | **4.35** | 22.26 |
| Latency on Tecnick | 61.01 | 33.81 | **9.48** | 47.74 |

Table 3: Discretization latency with different approaches (unit: microsecond).

Table 3 shows the inference latency when adopting comparison-based discretization used by Sun et al. (2021) and our proposed calculation-based discretization. All the results are tested on NVIDIA GTX 1060. To evaluate the comparison-based approach, we refer to popular CompressAI (Bégaint et al., 2020) implementation[2] and also test another more efficient vectorized algorithm (described in Appendix E.3). The inference latency of hyper synthesis is also reported as a reference. The results on Kodak (resolution: $512 \times 768$ px) prove that the directly calculated binary logarithm is more efficient, strongly suppressing the speed bottleneck of STD discretization. And the results on larger $1200 \times 1200$ px Tecnick images indicate that this improvement can be more significant when compressing high-resolution images.

## 6 DISCUSSION

The mature investigation on general model quantization provides free lunch to us for establishing a cross-platform consistent entropy estimation approach, which is essential to practical learned image compression. In this paper, we experimentally prove that the non-consistency issue of state-of-the-art learned image compression architectures can reduce to an integer-arithmetic-only model quantization problem. With a standard PTQ scheme, we achieve deterministic compression models which have almost the same compression performance as their pre-quantized full-precision versions. This result is encouraging. Furthermore, we improve the parameter discretization and extend it to fit GMM entropy model. In the future, we will further delve into practical learned image compression by extending the rate-distortion tradeoff to rate-distortion-speed tradeoff.

---

[2]`https://github.com/InterDigitalInc/CompressAI/blob/v1.1.8/compressai/entropy_models/entropy_models.py#L653-L658`

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

# A  DETAILED DESCRIBTION OF PTQ

## A.1  PER-CHANNEL WEIGHT QUANTIZATION

Weights of convolution filters may have various value ranges. Because of this imbalance, using the same quantization scale $s_{\mathbf{w}}$ for all filters may cause large rounding error in filters with small ranges and clipping error in filters with large ranges. Therefore, per-channel weight quantization is proposed (Krishnamoorthi, 2018; Li et al., 2019). To per-channel quantize a weight tensor $\mathbf{W} = [\mathbf{w}_1, \mathbf{w}_2, \dots, \mathbf{w}_c]$ with $c$ filter channels, we introduce $c$ scales $s_{\mathbf{w}_1}, \dots, s_{\mathbf{w}_c}$ and separately quantize each filter:

$$\hat{\mathbf{w}}_i = s_{\mathbf{w}_i} \mathbf{q}_{\mathbf{w}_i}, \quad i = 1, 2, \dots, c \tag{14}$$

where the scale factors $s_{\mathbf{w}_i}$ can be merged into later requantization operation.

Note that, we do not recommend to conduct this per-channel quantization on activation, as it will introduce extra calculation to rescale the multiplication results before convolution accumulation. Therefore, we only per-channel quantize the weights while keep activation per-tensor quantized, as Nagel et al. (2021) suggests.

## A.2  REQUANTIZATION

Consider a convolutional or fully connected layer with input $\mathbf{v}$, weights $\mathbf{W}$ and activation function $h(\cdot)$. The output activated vector $\mathbf{u}$ is:

$$\mathbf{u} = h(\mathbf{W}\mathbf{v}) \tag{15}$$

Provided that we have quantized the weights and input vector with scale factors $s_{\mathbf{W}}$ and $s_{\mathbf{v}}$ respectively, the corresponding quantized integer representation $\mathbf{Q}_{\mathbf{W}}$ and $\mathbf{q}_{\mathbf{v}}$ are:

$$\mathbf{Q}_{\mathbf{W}} = \text{clip}\left(\left\lceil (s_{\mathbf{W}}^{-1}\mathbf{W}) \right\rfloor\right) \tag{16}$$

$$\mathbf{q}_{\mathbf{v}} = \text{clip}\left(\left\lceil (s_{\mathbf{v}}^{-1}\mathbf{v}) \right\rfloor\right) \tag{17}$$

and the corresponding dequantization results are:

$$\hat{\mathbf{W}} = s_{\mathbf{W}}\mathbf{Q}_{\mathbf{W}} \tag{18}$$

$$\hat{\mathbf{v}} = s_{\mathbf{v}}\mathbf{q}_{\mathbf{v}} \tag{19}$$

Thus, the activation $\hat{\mathbf{u}}$ calculated from quantized weights and input is:

$$\hat{\mathbf{u}} = h(\hat{\mathbf{W}}\hat{\mathbf{v}}) = h(s_{\mathbf{W}}s_{\mathbf{v}}\mathbf{Q}_{\mathbf{W}}\mathbf{q}_{\mathbf{v}}) \tag{20}$$

Usually, we adopt ReLU or Leaky ReLU as the activation $h(\cdot)$. They are segmented linear functions with scaling invariance, *i.e.* scaling the input of $h(\cdot)$ with a factor $s$ is equivalence to scaling the output with $s$:

$$h(s\mathbf{v}) = s \cdot h(\mathbf{v}) \tag{21}$$

Therefore, we can move the scalar $s_{\mathbf{W}}s_{\mathbf{v}}$ in eq. 20 outside the activation function:

$$\hat{\mathbf{u}} = s_{\mathbf{W}}s_{\mathbf{v}}h(\mathbf{Q}_{\mathbf{W}}\mathbf{q}_{\mathbf{v}}) \tag{22}$$

Now $\mathbf{Q}_{\mathbf{W}}\mathbf{q}_{\mathbf{v}}$ is an integer matrix multiplication, outputting a 32-bit integer vector. Also, the activation function $h(\cdot)$ can be formulated as integer-arithmetic-only. Thus, the activation $h(\mathbf{Q}_{\mathbf{W}}\mathbf{q}_{\mathbf{v}})$ is a 32-bit integer vector, named $\mathbf{q}_{\mathbf{u}}$ by us. Now $\hat{\mathbf{u}}$ is a 32-bit fix-point number with integer representation $\mathbf{q}_{\mathbf{u}}$ and quantization scale factor $s_{\mathbf{W}}s_{\mathbf{v}}$:

$$\begin{aligned} \mathbf{q}_{\mathbf{u}} &= h(\mathbf{Q}_{\mathbf{W}}\mathbf{q}_{\mathbf{v}}) \\ \hat{\mathbf{u}} &= s_{\mathbf{W}}s_{\mathbf{v}}\mathbf{q}_{\mathbf{u}} \end{aligned} \tag{23}$$

Notice that, after each layer, the output should be quantized to 8 bits again, or the matrix multiplication in the next layer cannot be conducted using 8-bit integer arithmetic. Therefore, we quantize $\hat{\mathbf{u}}^{(\ell)}$ with scale factor $s_{\mathbf{v}}^{(\ell)+1}$ (note that $\mathbf{v}^{(\ell+1)} = \hat{\mathbf{u}}^{(\ell)}$):

$$\begin{aligned} \mathbf{q}_{\mathbf{v}}^{(\ell+1)} &= \text{clip}\left(\left\lceil \frac{1}{s_{\mathbf{v}}^{(\ell+1)}}\mathbf{v}^{(\ell+1)} \right\rfloor\right), \quad (\text{quantize } \mathbf{v}^{(\ell+1)}) \\ &= \text{clip}\left(\left\lceil \frac{1}{s_{\mathbf{v}}^{(\ell+1)}}\hat{\mathbf{u}}^{(\ell)} \right\rfloor\right), \quad (\mathbf{v}^{(\ell+1)} = \hat{\mathbf{u}}^{(\ell)}) \\ &= \text{clip}\left(\left\lceil \frac{s_{\mathbf{W}}^{(\ell)}s_{\mathbf{v}}^{(\ell)}}{s_{\mathbf{v}}^{(\ell+1)}}\mathbf{q}_{\mathbf{u}}^{(\ell)} \right\rfloor\right), \quad (\text{eq. 23}) \end{aligned} \tag{24}$$

Following Jacob et al. (2018), we fuse the division of scale factors to $m$, which is called the requantization scale factor:

$$m^{(\ell)} = \frac{s_{\mathbf{W}}^{(\ell)} s_{\mathbf{v}}^{(\ell)}}{s_{\mathbf{v}}^{(\ell+1)}} \tag{25}$$

Introducing $m$ to eq. 24, finally we obtain the requantization formula:

$$\mathbf{q}_{\mathbf{v}}^{(\ell+1)} = \text{clip}\left(\left\lceil m^{(\ell)} \mathbf{q}_{\mathbf{u}}^{(\ell)} \right\rfloor\right) \tag{26}$$

### A.3 FOLDING LEAKY RELU TO CONDITIONED DYADIC REQUANTIZATION

Leaky ReLU is frequently adopted in presently state-of-the-art compression models. For instance, Minnen et al. (2018) introduces it in hyper analyzer and synthesizer, Cheng et al. (2020) uses it as activation functions of the parameter network. Different from ReLU which can be simply seen as a conditioned clipping operator which is deterministic, Leaky ReLU involves floating point multiplication.

Considering Leaky ReLU with a negative slope $\alpha$ applied on each quantized scalar $q_{\mathbf{u}} \in \mathbf{q}_{\mathbf{u}}$, provided the requantization factor is $m$, the requantization result $q_{\mathbf{v}}$ is:

$$q_{\mathbf{v}}^{(\ell+1)} = \begin{cases} \text{clip}\left(\left\lceil m^{(\ell)}(q_{\mathbf{u}}^{(\ell)} + p_{\mathbf{u}}^{(\ell)}) \right\rfloor\right), & q_{\mathbf{u}}^{(\ell)} \geq 0 \\ \text{clip}\left(\left\lceil \alpha m^{(\ell)}(q_{\mathbf{u}}^{(\ell)} + p_{\mathbf{u}}^{(\ell)}) \right\rfloor\right), & q_{\mathbf{u}}^{(\ell)} < 0 \end{cases} \tag{27}$$

where $q_{\mathbf{u}}$ is a signed integer and checking its sign is painless. For non-negative and negative cumulative values, the above operation can be separately performed as two branches of requantization with different factors $m$ and $\alpha m$. Hence the Leaky ReLU can be fused into the requantization in a vectorized manner. We implement the fused activation layer as a conditioned dyadic multiplication, with scaling factor $m$ for non-negative $q_{\mathbf{u}}$ and $\alpha m$ for negative $q_{\mathbf{u}}$.

## B EQUATION DERIVATIONS

### B.1 DERIVATIONS OF EQUATION 9

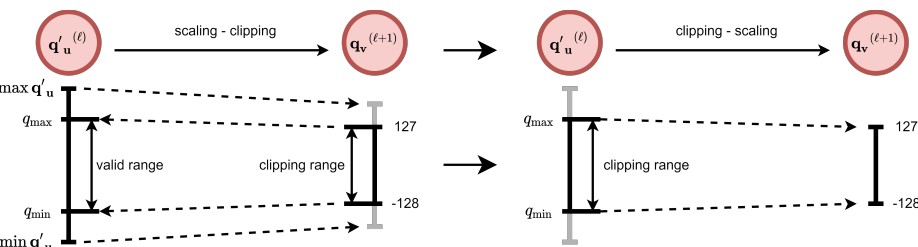

Figure 6: Comparison between scaling-clipping requantization and clipping-scaling requantization. By computing the bounds of valid range, we can conduct the clipping operation before scaling with factor $m$.

Revisit eq. 8 in the main body:

$$q_{\max}^{(\ell)} = \left\lfloor \frac{2^{B-1} - 1}{m^{(\ell)}} \right\rfloor$$

$$q_{\min}^{(\ell)} = \left\lceil \frac{-2^{B-1}}{m^{(\ell)}} \right\rceil$$

The bounds $q_{\max}, q_{\min}$ describe the valid value range of the activation $\mathbf{q}'_{\mathbf{u}}$ biased by the zero-point $p_{\mathbf{u}}$. After the clipping operation, all of the clipped value $\mathbf{q}''_{\mathbf{u}}$ will be in this range:

$$\mathbf{q}''_{\mathbf{u}} = \text{clip}\left(\mathbf{q}'_{\mathbf{u}}, q_{\min}, q_{\max}\right)$$
$$\text{s.t. } \forall q \in \mathbf{q}''_{\mathbf{u}}, \ -2^{B-1} \leq mq_{\min} \leq mq \leq mq_{\max} \leq 2^{B-1} - 1 \tag{28}$$

To perform a dyadic requantization, we want to find out proper integers $m_0, n$ subject to

$$\forall q \in \mathbf{q}''_{\mathbf{u}}, \; -2^{B-1} \le \left\lceil \frac{m_0 q}{2^n} \right\rceil \le 2^{B-1} - 1 \tag{29}$$

We let $m_0$ become a function of $n$:

$$m_0 = \lfloor 2^n m \rfloor \tag{30}$$

Notice that $m_0$ is non-negative and $n, m$ is positive. For each $q \in \mathbf{q}''_{\mathbf{u}}$,

$$\left\lceil \frac{m_0 q}{2^n} \right\rceil \le \left\lceil \frac{m_0 q_{\max}}{2^n} \right\rceil = \left\lceil \frac{\lfloor 2^n m \rfloor}{2^n} q_{\max} \right\rceil \le \lceil m q_{\max} \rceil \le \left\lceil 2^{B-1} - 1 \right\rceil = 2^{B-1} - 1$$
$$\left\lceil \frac{m_0 q}{2^n} \right\rceil \ge \left\lceil \frac{m_0 q_{\min}}{2^n} \right\rceil = \left\lceil \frac{\lfloor 2^n m \rfloor}{2^n} q_{\min} \right\rceil \ge \lceil m q_{\min} \rceil \ge \left\lceil -2^{B-1} \right\rceil = -2^{B-1} \tag{31}$$

which satisfies the condition in eq. 29.

Therefore, we only need to determine a proper value of $n$ following two conditions:

1. The multiplication overflow from $m_0 \mathbf{q}''_{\mathbf{u}}$ should be avoided. As $m_0$ is a function driven by $n$ now, this condition implicitly define an upper bound of $n$.

2. $n$ should be set as large as possible, to suppress the rounding error introduced by requantization.

To avoid the 32-bit multiplication overflow, we constrain the magnitude of $m_0 \mathbf{q}''_{\mathbf{u}}$ subject to:

$$\forall q \in \mathbf{q}''_{\mathbf{u}}, \; -2^{31} \le m_0 q \le 2^{31} - 1 \tag{32}$$

Introducing $m_0 = \lfloor 2^n m \rfloor$, we obtain the conditions:

$$\lfloor 2^n m \rfloor \, q_{\max} \le 2^{31} - 1 \tag{33}$$
$$\lfloor 2^n m \rfloor \, q_{\min} \ge -2^{31} \tag{34}$$

Further introducing the definition of $q_{\max}$ and $q_{\min}$ in eq. 8, we have:

$$\lfloor 2^n m \rfloor \le \frac{2^{31} - 1}{q_{\max}} = \frac{2^{31} - 1}{\left\lfloor \frac{2^{B-1} - 1}{m} \right\rfloor}$$
$$< \frac{2^{31}}{\left\lfloor \frac{2^{B-1} - 1}{m} \right\rfloor} \tag{35}$$

$$\lfloor 2^n m \rfloor \le \frac{-2^{31}}{q_{\min}} = \frac{-2^{31}}{\left\lceil \frac{-2^{B-1}}{m} \right\rceil} = \frac{2^{31}}{\left\lfloor \frac{2^{B-1}}{m} \right\rfloor}$$
$$\le \frac{2^{31}}{\left\lfloor \frac{2^{B-1} - 1}{m} \right\rfloor} \tag{36}$$

Since $m$ is positive, we have $\lfloor 2^n m \rfloor \le 2^n m$ and $\frac{2^{31}}{\frac{2^{B-1}}{m}} \le \frac{2^{31}}{\left\lfloor \frac{2^{B-1}}{m} \right\rfloor} \le \frac{2^{31}}{\left\lfloor \frac{2^{B-1} - 1}{m} \right\rfloor}$. Hence we relax the condition to let $n$ subject to:

$$2^n m \le \frac{2^{31}}{\frac{2^{B-1}}{m}} \tag{37}$$

We get:

$$2^n \le 2^{32-B} \tag{38}$$

so that we let $n = 32 - B$. It is almost the largest value of $n$ subject to the condition in eq. 29 and eq. 32, avoiding the multiplication $m_0 \mathbf{q}''_{\mathbf{u}}$ overflow.

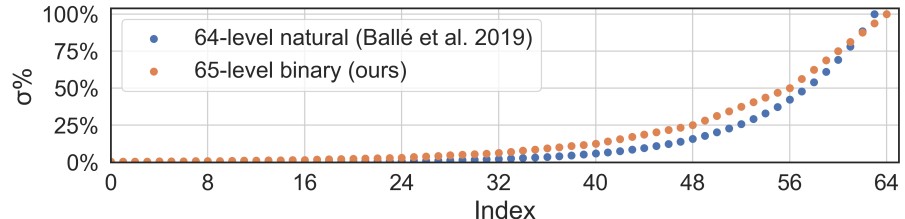

Figure 7: Comparison of our binary logarithm discretization and natural logarithm discretization proposed in Ballé et al. (2019). The Y-axis is the percent of discretized standard deviation introduced for meaningful comparison: $\sigma\% = \frac{\hat{\sigma}}{\sigma_{\max}} \times 100\%$ with $\sigma_{\max} = 256$ for the natural logarithm one and $\sigma_{\max} = 32$ for ours.

### B.2   DERIVATIONS OF EQUATION 12

The 65 level STD discretization is derived from a linear interpolation. Revisit the definition of major levels in eq. 11 and consider $\sigma = sq$:

$$\hat{i} = \lfloor \log_2(q) \rfloor - 3$$

$$\hat{\sigma}_{\mathrm{major}} = \sigma_{\min}(\exp2(\hat{i}))$$

Since $q$ has been clipped to $[8, 2048]$ (corresponding to dequantized $\sigma \in [0.125, 32]$ with quantization step $2^{-6}$), $\hat{i} \in \{0, 1, \dots 8\}$ and $\hat{\sigma}_{\mathrm{major}} \in \{2^{-3}, 2^{-2}, \dots, 2^5\}$.

The discretization will map all $\sigma \in [0.125, 0.25)$ to $0.125$ and $\sigma \in [0.25, 0.5)$ to $0.25$, which is too sparse and will result in large error, significantly hurting the compression performance. Thus, we interpolate minor values between two adjacent major values $\hat{i}$ and $\hat{i} + 1$.

Let $\hat{\sigma}_{\hat{i}} = \sigma_{\min}(\exp2(\hat{i}))$ denotes the STD corresponding to index $\hat{i}$. And similarly let $\hat{\sigma}_{\hat{i}+1}$ corresponds to $\hat{i} + 1$. We linearly insert 7 minor levels in $[\hat{i}, \hat{i} + 1]$ with the step size $\Delta_{\mathrm{minor}}$:

$$
\begin{aligned}
\Delta_{\mathrm{minor}} &= \frac{\hat{\sigma}_{\hat{i}+1} - \hat{\sigma}_{\hat{i}}}{8} \\
&= \frac{1}{8}\sigma_{\min}(\exp2(\hat{i} + 1) - \exp2(\hat{i})) \\
&= \frac{1}{8}\sigma_{\min}\exp2(\hat{i})
\end{aligned}
\tag{39}
$$

and the minor index is:

$$
\begin{aligned}
\hat{j} &= \left\lceil \Delta_{\mathrm{minor}}^{-1}(\sigma - \hat{\sigma}_{\hat{i}}) \right\rceil \\
&= \left\lceil 8\sigma_{\min}^{-1}\exp2(-\hat{i})(\sigma - \hat{\sigma}_{\hat{i}}) \right\rceil \\
&= \left\lceil 8\exp2(-\hat{i})(\frac{q}{8} - \exp2(\hat{i})) \right\rceil \\
&= \left\lceil \frac{(q - \exp2(\hat{i} + 3))}{\exp2(\hat{i})} \right\rceil
\end{aligned}
\tag{40}
$$

where $\hat{\sigma}_{\hat{i}} \leq \sigma < \hat{\sigma}_{\hat{i}+1}$ and $\sigma = 2^{-6}q$. So $\hat{i} \leq \lfloor \log_2(q) \rfloor - 3 < \hat{i} + 1$, thus, $\hat{i} = \lfloor \log_2(q) \rfloor - 3$. Introduce it to the above equation, we have:

$$\hat{j} = \left\lceil \frac{(q - \exp2(\lfloor \log_2(q) \rfloor))}{\exp2(\lfloor \log_2(q) \rfloor - 3)} \right\rceil \tag{41}$$

Thus, the STD $\sigma$ will be discretized to:

$$\hat{\sigma} = \hat{\sigma}_{\hat{i}} + j\Delta_{\mathrm{minor}} \tag{42}$$

And the interpolated levels are $\hat{\sigma}_{\hat{i}} + \Delta_{\mathrm{minor}}, \hat{\sigma}_{\hat{i}} + 2\Delta_{\mathrm{minor}}, \dots, \hat{\sigma}_{\hat{i}} + 7\Delta_{\mathrm{minor}}$. When $\hat{\sigma}_{\hat{i}} + 7\Delta_{\mathrm{minor}} < \sigma < \hat{\sigma}_{\hat{i}} + 8\Delta_{\mathrm{minor}} = \hat{\sigma}_{\hat{i}+1}$, the STD $\sigma$ will be mapped to the next major level $\hat{\sigma}_{\hat{i}+1}$. When it occurs, the corresponding minor index $\hat{j} = 8$.

## C    EXPERIMENT: DETAILED SETTINGS AND MORE RESULTS

### C.1    TRAINING FLOATING-POINT MODELS

To evaluate the proposed PTQ-based scheme, we should first obtain the full-precision models trained on floating-point numbers. In this section we will report our detailed training settings for reproducibility.

We draw 8000 images with largest resolution from ImageNet (Deng et al., 2009) to construct the training dataset. The training set is shared to train all reported models. Before training we apply the same preprocessing as Ballé et al. (2017) on all data by downsamping and disturbing the input images. During training, we randomly crop each image to $256 \times 256$ px in each iteration. We adopt a batch-size of 16 and a initial learning rate of 1e-4 in all training. To optimize the parameters, we use Adam optimizer with $\beta_1 = 0.9, \beta_2 = 0.999$. To cover a range of rate–distortion tradeoffs, 6 different values of $\lambda$ are chosen from $\{0.0016, 0.0032, 0.0075, 0.015, 0.03, 0.45\}$ for MSE-optimization, following previous works (Cheng et al., 2020; He et al., 2021). On all models, we adopt spectral Adam optimization (Sadam, Ballé (2018)) to improve the training stability. For each model architecture, we particularly adjust the training setting according to previous suggestions:

- **Ballé et al. (2018)**. For each $\lambda$ we train the corresponding model 2000 epochs. As suggested by the authors, we set a so-called bottleneck with $N = 128, M = 192$ when using the 3 lower $\lambda$ and $N = 192, M = 320$ when using the others.

- **Minnen et al. (2018)**. Following the suggestion from the authors[3], We train 6000 epochs. We choose $N$ and $M$ dependent on $\lambda$, with $N = 128$ and $M = 192$ for the 3 lower $\lambda$, and $N = 192$ and $M = 320$ for the 3 higher ones. We decay the learning rate to 5e-5 after training 3000 epochs.

- **Cheng et al. (2020)**. As recommended by He et al. (2021), we train 6000 epochs to achieve well-optimized performance of Cheng et al. (2020). $N$ is set as 128 for the 3 lower-rate models, and set as 192 for the 3 higher-rate models. We decay the learning rate to 5e-5 after training 3000 epochs.

### C.2    EVALUATION

Following prior works, we use Kodak (Kodak, 1993) and Tecnick (Asuni & Giachetti, 2014) as our test dataset. The results of image compression are shown as rate-distortion curves. Following previous works, we use PSNR and MS-SSIM (Wang et al., 2003) as the distortion metric to measure the reconstruction quality. We calculate bits-per-pixel (BPP) as the rate metric to measure the compressed file size.

### C.3    CALIBRATION

We use a calibration dataset to calculate the activation quantization steps. To avoid data leakage, we use DIV2K (Agustsson & Timofte (2017)) as the calibration dataset. We center-crop each one of 900 images in DIV2K to $256 \times 256$ pixels and feed them into the quantization pipeline with a batch-size of 32.

Recommended by Nagel et al. (2021), we search the weight quantization step using a grid search method. Given full-precision weight vector $\mathbf{w}$, which is represented by floating point, we minimize its reconstruction mean-square-error to find the optimal quantization step $s_\mathbf{w}$:

$$s_\mathbf{w}^* = \arg\min_{s_\mathbf{w}} \|\hat{\mathbf{w}} - \mathbf{w}\|_2 \quad \text{s.t. } s_\mathbf{w} \in \{s_1, s_2, \dots, s_N\} \tag{43}$$

where $N$ is the volume of the search space and $\hat{\mathbf{w}}$ is the reconstructed vector $\mathbf{w}$ which is quantized with factor $s_\mathbf{w}$. This approach can be directly applied on each convolution filter $\mathbf{w}_i$ to conduct the per-channel weight quantization.

---

[3] https://groups.google.com/g/tensorflow-compression/c/LQtTAo6l26U/m/cD4ZzmJUAgAJ

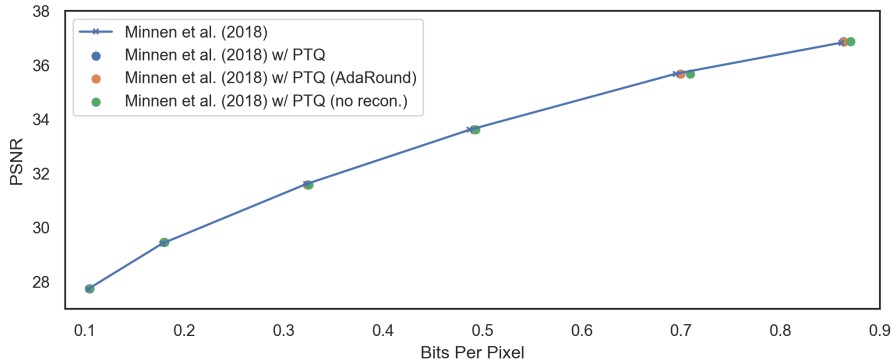

Figure 8: Comparison of quantized Minnen et al. (2018) w/ and w/o Brecq weight rounding reconstruction. The results are evaluated on Kodak.

To obtain the full-precision activation, we feed a batch of calibration data to the model before quantization and store the floating-point activation outputs. And the range of saved full-precision activation $\mathbf{u}$ will be used to obtain step $s_{\mathbf{u}}$ and zero-point $z_{\mathbf{u}}$ with Min-Max approach:

$$s_{\mathbf{u}} = \frac{(\max \mathbf{u} - \min \mathbf{u})}{2^B - 1} \tag{44}$$

$$z_{\mathbf{u}} = -\left\lceil s_{\mathbf{u}}^{-1} \min \mathbf{u} \right\rceil \tag{45}$$

## C.4 EFFECTS OF WEIGHT ROUNDING RECONSTRUCTION.

| **Method** | w/o reconstruction | AdaRound (Nagel et al., 2020) | Brecq (Li et al., 2020) |
|---|---|---|---|
| Minnen et al. (2018) | 1.329% | 0.741% | 0.663% |
| Cheng et al. (2020) | 0.607% | 0.415% | 0.416% |

Table 4: Relative BPP increment on Kodak when using (or not) different weight rounding reconstruction approaches, compared with full-precision model.

We evaluate quantization results with and without weight rounding reconstruction, *i.e.* AdaRound (Nagel et al., 2020) or Brecq (Li et al., 2020). Shown in Figure 8 and Table 4, the direct quantization without per-layer reconstruction has achieved almost no performance loss at lower bit rates (BPP $< 0.4$). The rate increment at higher bit rates can be compensated by Brecq. Brecq is directly performed on weight offline, it is relatively costless and we adopt it to establish our PTQ baseline.

We implement Brecq according to the open-source code provided by the authors. We view the hyper synthesizer, context model and parameter network as three blocks to perform weight reconstruction, *i.e.* we adopt the per-block reconstruction mentioned in Li et al. (2020). We use MSE as the object instead of FIM for simplicity. We use a batch-size of 32, and learning rate of 1e-3 with Adam to perform the gradient descent, tuning each block for 20000 iterations.

## C.5 MORE RATE-DISTORTION RESULTS

For completeness, following existing work (Ballé et al., 2018; Minnen et al., 2018; Cheng et al., 2020; He et al., 2021), we further report RD performance evaluated on Tecnick (Figure 10) and RD performance of models optimized on MS-SSIM (Figure 9). After the quantization, the models keep comparable performance.

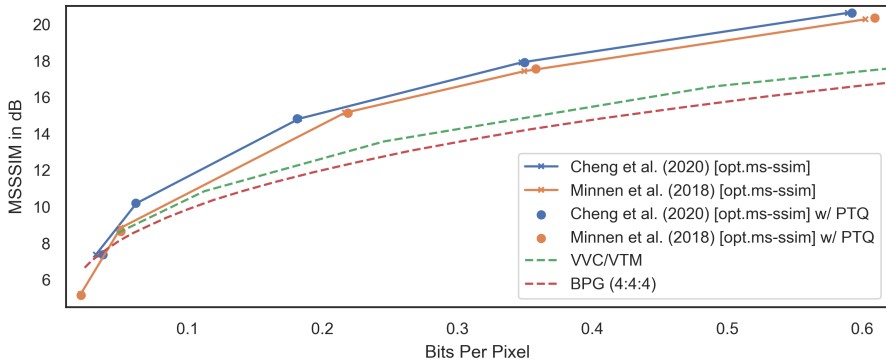

Figure 9: Influence of quantization to models with MS-SSIM optimization. Evaluated on Kodak.

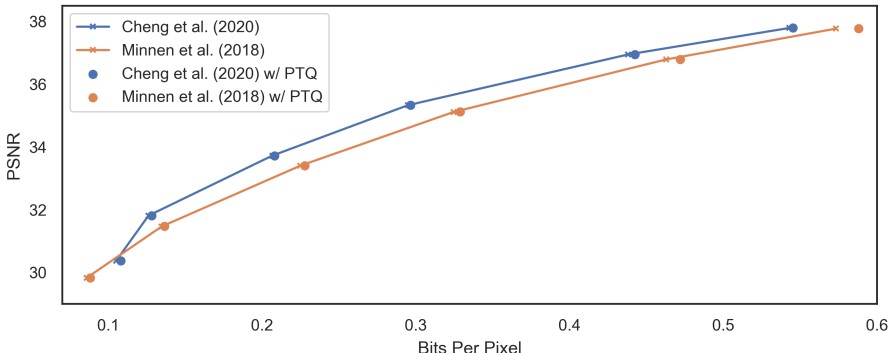

Figure 10: The same models as Figure 5(a) evaluated on Tecnick.

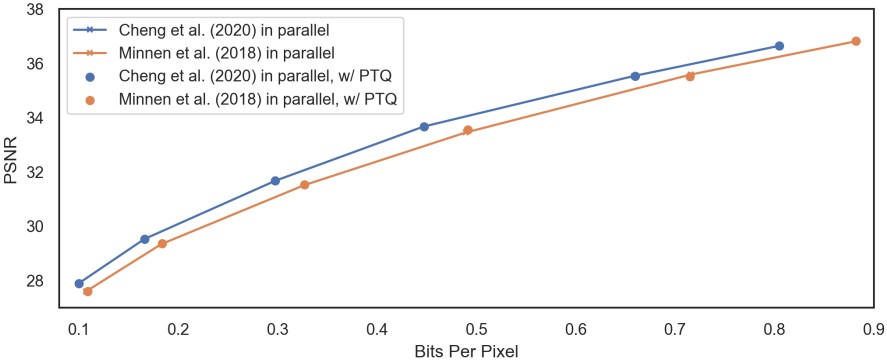

Figure 11: RD curves of models with parallel context models, quantized or not. Evaluated on Kodak.

### C.6 ORTHOGONALITY WITH PARALLEL CONTEXT MODEL

An issue of learned image compression with context models is the inefficiency of serial decoding. He et al. (2021) proposes to address this problem by developing a checkerboard-shaped parallel context modeling scheme, instead of the original serial autoregressive method. We also investigate the quantization of Minnen et al. (2018) and Cheng et al. (2020) with this parallel context adaption. As shown in Figure 11, PTQ still performs well without hurting the RD performance, making the learned image compression pragmatic.

## D LEARNED IMAGE COMPRESSION

### D.1 OVERVIEW

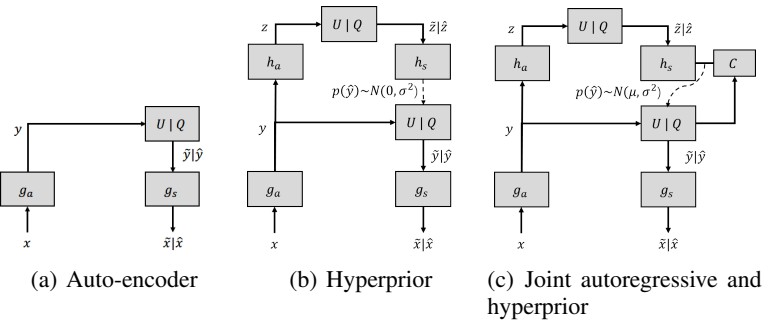

(a) Auto-encoder        (b) Hyperprior        (c) Joint autoregressive and hyperprior

Figure 12: Operation diagrams of different learned image compression architectures. (a) an auto-encoder like model (Ballé et al., 2017) (b) Scale hyperprior model. $h_a$ and $h_s$ represent hyper analysis and synthesis while $\mathbf{z}$ denotes the hyperprior (Ballé et al., 2018). (c) Joint autoregressive and hyperprior model. $C$ denotes the context model (Minnen et al., 2018).

As shown in Figure 12(a), the widely adopted framework of auto-encoder like learned image compression proposed by Ballé et al. (2017) is similar to transform coding (Goyal, 2001). During encoding, the input image $\mathbf{x}$ is mapped to the latent representation $\mathbf{y}$ through a parametric analysis transform $g_a(\cdot)$. After the round-to-nearest quantization, the quantized $\hat{\mathbf{y}}$ is used as coding symbols, which will be encoded into the bitstream later. To losslessly encode $\hat{\mathbf{y}}$ into a short enough bitstream, an entropy model $p_{\hat{\mathbf{y}}}(\hat{\mathbf{y}})$ is introduced to fit the probability mass of $\hat{\mathbf{y}}$, so that we can use entropy encoders. With the same entropy model and an entropy decoder, $\hat{\mathbf{y}}$ is decoded from the compressed data stream, and fed to a learned synthesis transform $g_s(\cdot)$ to produce the reconstructed image $\hat{\mathbf{x}}$.

To train the auto-encoder like model with stochastic gradient descent, a uniform noise estimator is introduced to replace the rounding operation (whose gradient is zero almost everywhere) during training. It produces noisy vector $\tilde{\mathbf{y}} = \mathbf{y} + \mathbf{r}$ where $\mathbf{r} \sim U(-0.5, 0.5)$, to approximate the quantized $\hat{\mathbf{y}}$ during training. Thus, we can fit a continuous distribution $p_{\tilde{\mathbf{y}}}(\tilde{\mathbf{y}})$, which is modeled as a segmented linear parametric model in Ballé et al. (2017). Hereinafter we always use $\hat{\mathbf{x}}$, $\hat{\mathbf{y}}$ and $\hat{\mathbf{z}}$ to represent $\tilde{\mathbf{x}}|\hat{\mathbf{x}}$, $\tilde{\mathbf{y}}|\hat{\mathbf{y}}$ and $\tilde{\mathbf{z}}|\hat{\mathbf{z}}$ for simplicity.

By supervising the entropy, the training of learned image compression model is formulated as rate-distortion optimization. The loss function can be expressed as:

$$\begin{aligned} \mathcal{L} &= R + \lambda \cdot D \\ &= \mathbb{E}[-log_2 p_{\hat{\mathbf{y}}}(\hat{\mathbf{y}})] + \lambda \cdot \mathbb{E}[d(\mathbf{x}, \hat{\mathbf{x}})] \end{aligned} \tag{46}$$

where $\lambda$ is the coefficient to trade-off rate and distortion. $d(\mathbf{x}, \hat{\mathbf{x}})$ denotes the distortion between the original image $\mathbf{x}$ and the reconstructed image $\hat{\mathbf{x}}$ where mean squared error (MSE) is the most common choice.

Shown in Figure 12(b), in Ballé et al. (2018), the entropy of $\hat{\mathbf{y}}$ is estimated using a conditioned Gaussian scale model (GSM) by introducing hyper latent $\mathbf{z}$ as side information:

$$
\begin{aligned}
p_{\hat{\mathbf{y}}|\hat{\mathbf{z}}}(\hat{y}_i|\hat{\mathbf{z}}) &= \left[\mathcal{N}(0, \sigma_i^2) * U(-0.5, 0.5)\right](\hat{y}_i) \\
&= \int_{\hat{y}_i-0.5}^{\hat{y}_i+0.5} \mathcal{N}(0, \sigma_i^2)(y)dy \\
p_{\hat{\mathbf{y}}|\hat{\mathbf{z}}}(\hat{\mathbf{y}}|\hat{\mathbf{z}}) &= \prod_i p_{\hat{\mathbf{y}}|\hat{\mathbf{z}}}(\hat{y}_i|\hat{\mathbf{z}})
\end{aligned}
\tag{47}
$$

where $\mathbf{z}$ is the hyperprior calculated from $\mathbf{y}$ with a hyper analyzer $h_a(\cdot)$ and each $\sigma_i$ is inferred from $\hat{\mathbf{z}}$ by a hyper synthesizer $h_s(\cdot)$. By encoding and decoding $\hat{\mathbf{z}}$ independently, this hyperprior is used as side-information to perform a forward-adaptive coding, conditioning on which the entropy of each $\hat{y}_i$ can be better modeled.

Shown in Figure 12(c), in Minnen et al. (2018) backward-adaption is also introduced, forming a joint autoregressive and hyperprior coding scheme. Provided that the currently en/de-coding symbol is $\hat{y}_i$, we can predict it with the already visible symbols $\hat{\mathbf{y}}_{j<i}$ which have been en/de-coded before $\hat{y}_i$. Further adopting a mean-scale Gaussian entropy model which extends GSM, we can achieve a more flexible entropy model:

$$
\begin{aligned}
p_{\hat{\mathbf{y}}|\hat{\mathbf{z}},\hat{\mathbf{y}}_{j<i}}(\hat{y}_i|\hat{\mathbf{z}}, \hat{\mathbf{y}}_{j<i}) &= \left[\mathcal{N}(\mu_i, \sigma_i^2) * U(-0.5, 0.5)\right](\hat{y}_i) \\
&= \int_{\hat{y}_i-0.5}^{\hat{y}_i+0.5} \mathcal{N}(\mu_i, \sigma_i^2)(y)dy
\end{aligned}
\tag{48}
$$

where element-wise entropy parameters $\mu_i$ and $\sigma_i$ are jointly predicted from context model, hyper synthesizer and parameter network as highlighted in Figure 2. Cheng et al. (2020) further introduces Gaussian mixture model (GMM) with $K$ components as the entropy model:

$$
p_{\hat{\mathbf{y}}|\hat{\mathbf{z}},\hat{\mathbf{y}}_{j<i}}(\hat{y}_i|\hat{\mathbf{z}}, \hat{\mathbf{y}}_{j<i}) = \sum_{k=1}^{K} \int_{\hat{y}_i-0.5}^{\hat{y}_i+0.5} \pi^{(k)} \mathcal{N}(\mu_i^{(k)}, \sigma_i^{2(k)})(y)dy
\tag{49}
$$

subject to $\sum_{k=1}^{K} \pi^{(k)} = 1$. The entropy parameters $\pi_i^{(k)}, \mu_i^{(k)}, \sigma_i^{(k)}$ are still jointly predicted from context model, hyper synthesizer and parameter network. Recently, this GMM-based optimization is further promoted by introducing global context modeling and grouped context modeling (Guo et al., 2021), which outperforms VVC/VTM on both PSNR and MS-SSIM.

### D.2 ENTROPY CODING WITH ARITHMETIC CODERS AND THE INCONSISTENCY ISSUE

In learned image compression, we usually adopt arithmetic en/de-coder (AE/AD) or its variants, range coder (Martin, 1979) and asymmetric numeral systems (ANS), to compress/decompress the symbols $\hat{\mathbf{y}}$ to/from bitstream. Using AE as an example, during encoding AE requires both $\hat{\mathbf{y}}$ and its cumulative distribution function (CDF) $C_{\hat{\mathbf{y}}}$ as input. AE will allocate bits for each latent $\hat{y}_i$ according to the CDF. The same CDF should be fed to AD during decoding to correctly decompress $\hat{\mathbf{y}}$. This is exactly what we are investigating in this work: making the CDF estimated by entropy model consistent across different platforms to ensure the correct decoding of entropy coding.

Calculating CDF relies on the output of parameter network, which is continuous floating-point representation. Thus, the calculation of CDF is usually expensive and non-deterministic floating-point arithmetic. For instance, CDF for a discrete mean-scale Gaussian distribution (Ballé et al., 2018) is:

$$
C_{\hat{y}}(\hat{y}; \mu, \sigma) = \sum_{-\infty}^{\hat{y}} \int_{\hat{y}-0.5}^{\hat{y}+0.5} \mathcal{N}(\mu, \sigma^2)(\hat{y})d\hat{y} = \Phi\left(\frac{\hat{y} - \mu + 0.5}{\sigma}\right)
\tag{50}
$$

where $\hat{y}$ is one element in the quantized integer latent representations $\hat{\mathbf{y}}$ (we omit the subscript $i$ for simplicity). $\Phi(\cdot)$ is the CDF of standard Gaussian distribution and is often calculated from the standard Gaussian probability density function ndtr[4] or the error function erfc[5], which is often approx-

---

[4] https://github.com/tensorflow/probability/blob/v0.14.1/tensorflow_probability/python/distributions/normal.py#L199

[5] https://github.com/InterDigitalInc/CompressAI/blob/v1.1.8/compressai/entropy_models/entropy_models.py#L577

imated by interpolation algorithms. Therefore, even after quantizing the parameter network output: the mean $\mu$ and the standard deviation (STD) $\sigma$, the calculation of CDF is still non-deterministic.

Existing implementations pre-compute and store limited number of CDFs into look-up-tables (LUTs) to address this problem. By directly distributing the saved CDF tables across various platforms, the inconsistency in calculating CDF is eliminated. To conduct the LUT-based CDF query, the mean and STD should be discretized to limited sets of values $\{\hat{\mu}\}$ and $\{\hat{\sigma}\}$. Otherwise, the non-constrained parameters will generate infinite CDFs, which cannot be stored. We will discuss this discretization in the following section, Appendix D.3.

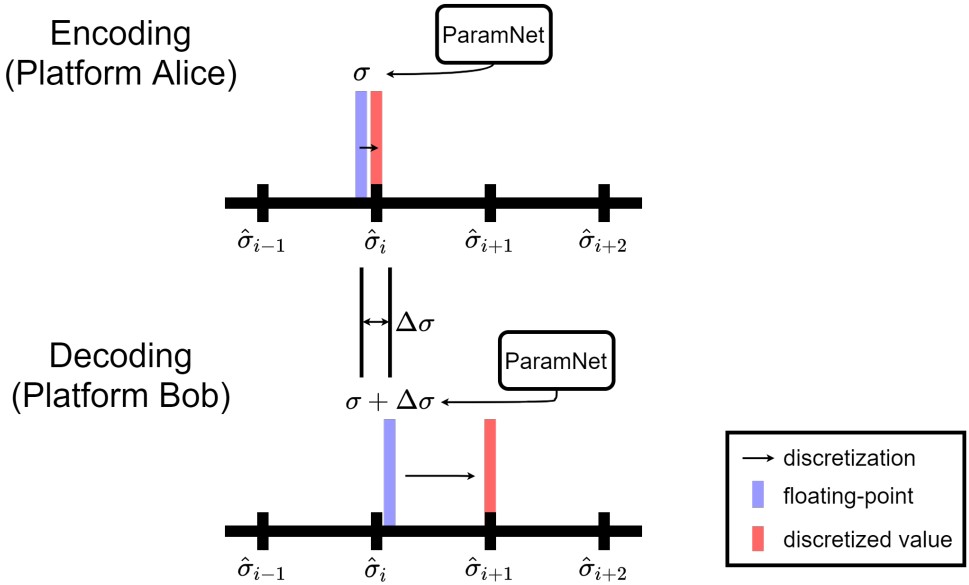

Figure 13: When the cross-platform inconsistency occurs, the discretized $\hat{\sigma}$ (and also $\hat{\mu}$) may become different between the sender Alice and the receiver Bob. Thus, the corresponding CDF index $i$ and $i + 1$ differ, which ends up with failed decoding.

When the calculation inconsistency occurs, the discretized parameters and their corresponding indexes may be different across encoder and decoder (Figure 13). When encoding symbol $\hat{y}$ with the inferred STD $\sigma$, for instance, Alice discretizes $\sigma$ to the $i$-th sampling point $\hat{\sigma}_i$. Then Alice will use the $i$-th CDF to allocate bits for $\hat{y}$ and encode it. After all the symbols have been encoded, Alice send the bitstream to Bob who will decode the symbols on another platform. Because of the non-determinism, Bob infers the entropy parameter of $\hat{y}$ as $\sigma + \Delta\sigma$ with error $\Delta\sigma$ different from Alice. Thus, the discretized parameter is $\hat{\sigma}_{i+1}$ with corresponding index $i + 1$. Then Bob tries to incorrectly decode $\hat{y}$ with the $(i + 1)$-th CDF instead of the $i$-th. Because the arithmetic coding is of a recursion form, the failed decoding of symbol $\hat{y}$ results in failed decoding of all the following symbols. It corrupts the decoded image, as shown in Figure 1.

### D.3 PARAMETER DISCRETIZATION FOR CDF INDEXING

As mentioned in Appendix D.2, we should discretize the entropy parameters for LUT query. Ballé et al. (2019) proposes to constrain each predicted $\sigma$ to values in a limited discrete set:

$$\Delta_\sigma = \frac{\log(\sigma_{\max}) - \log(\sigma_{\min})}{L - 1}$$
$$\hat{i}_\sigma = \left\lfloor \frac{\log(\sigma) - \log(\sigma_{\min})}{\Delta_\sigma} \right\rfloor \tag{51}$$
$$\hat{\sigma} = \exp\left(\hat{i}_\sigma \Delta_\sigma + \log(\sigma_{\min})\right)$$

where the lower and upper bounds $\sigma_{\min}, \sigma_{\max}$ are constant values. The input $\sigma$ should get clipped into range $[\sigma_{\min}, \sigma_{\max}]$ and the clipping is omitted in above formula for simplicity. The discretizing

level $L$ is set to 64. Since each index $\hat{i}_\sigma$ corresponds to a particular $\hat{\sigma}$ which will generate a CDF, we need to obtain the indexes in order to perform a LUT-based CDF query. In models using integer networks proposed in Ballé et al. (2019), the parameter network outputs the 6-bit integer index $\hat{i}_\sigma$ directly. But in uniform quantization-based approaches, the parameter network instead outputs dequantized fix-point $\sigma$. Thus, we need extra map the fix-bit output to the integer index of LUT. Sun et al. (2021) obtains $\hat{i}_\sigma$ by comparing the fix-point value with pre-computed discretized $\hat{\sigma}$.

The discretization of means has been addressed in Sun et al. (2021), with an observation that:

$$C_{\hat{y}}(\hat{y}; \mu, \sigma) = C_{\hat{y}}(\hat{y} - \lfloor \mu \rfloor; \mu - \lfloor \mu \rfloor, \sigma) \tag{52}$$

where $\mu - \lfloor \mu \rfloor \in [0, 1)$ is the decimal part of $\mu$. Thus, we can just uniformly discretize the decimal part of $\mu$ to $M$ levels and store a limited number of CDFs:

$$\hat{i}_\mu = \lceil (\mu - \lfloor \mu \rfloor)M \rfloor, \quad \hat{\mu} = \hat{i}_\mu M^{-1} + \lfloor \mu \rfloor \tag{53}$$

Therefore, we can obtain $LM$ pairs of $(\hat{i}_\mu, \hat{i}_\sigma)$, corresponding to $LM$ CDFs. For all CDFs, we pre-compute their function values on the input range $\{-R, -R+1, \ldots, 0, \ldots, R-1, R\}$ and save them as LUTs with lengths of $2R + 2$, where each table includes an extra element representing ending-of-bitstream.

## E  ALGORITHMS

### E.1  PARAMETER DISCRETIZATION

The discretization method described in eq. 12 is used to convert the 16-bit network output to CDF indexes. We give its element-wise description with pseudo-code in Algorithm 1.

---

**Algorithm 1:** Binary logarithm discretization with interpolation

**Input:** 16-bit integer $q$
**Output:** Corresponding CDF index $\hat{i}_\sigma$

```
/* Clip q with lower and upper bounds                                    */
```
1 **if** $q < 8$ **then**
2     $q := 8$            `// ` $\sigma_{\min} = 0.125$ `quantized by step size` $2^{-6}$
3 **else if** $q > 2048$ **then**
4     $q := 2048$          `// ` $\sigma_{\max} = 32$ `quantized by step size` $2^{-6}$
5 $b := \text{IntLog2}(q)$
6 $\hat{i} := b - 3$
7 $e_1 := 1 << \text{b}$
8 $e_2 := 1 << (\text{b} - 3)$
9 $\hat{j} := (q - e_1 + e_2 - 1)/e_2$    `// round-up integer-division by adding ` $e_2 - 1$
10 $\hat{i}_\sigma := 8 \times \hat{i} + \hat{j}$

---

### E.2  DETERMINISTIC EN/DE-CODING WITH GAUSSIAN MIXTURE MODEL

For simplicity, in this section we use $i_k$ to represent the joint outer index $\hat{i}_\mu L + \hat{i}_\sigma$ for $k_{th}$ Gaussian component. We use $\text{CDF}[\cdot]$ to denote the outer query by $i_k$ and $C_k[\cdot]$ to denote the inner query by $\hat{y} - \lfloor \mu_k \rfloor$. The output CDF value $c_{\hat{y}}$ in Algorithm 2 is the frequency cumulate below $\hat{y}$.

The encoding is described in Algorithm 3, where Line 2 denotes symbol $\hat{y}$ bigger than $(\max \lfloor \mu \rfloor) + R$ where the CDF value is $\text{CDF}_{\max}$, or smaller than $(\min \lfloor \mu \rfloor) - R$ where the CDF value is zero. In this case, Line 3 and 5 encode symbol $-R$ as a placeholder in AE (ArithmeticEnc(symbol, lower_cumulate, upper_cumulate, state)), while Line 4 encodes $\hat{y}$ using Golomb coding (GolombEnc(symbol, state). The decoding process is given in Algorithm 4, where the reverse process is performed.

### E.3 Our faster vectorized implementation of comparison-based discretization

Existing implementation of comparison-based discretization adopted by Sun et al. (2021) is for-loop manner, which is somewhat inefficient. For a fair comparison, we further provide a vectorized implementation. Its PyTorch code is like:

```
# initialization
scale_table = np.exp(np.linspace(np.log(0.11), np.log(256), 64))
scale_table = torch.tensor(scale_table[:-1]).cuda()
scale_table = scale_table[None, None, None, None, :]

# running
sigma_expand = sigma.unsqueeze(-1)
lut_index = (sigma_expand > scale_table).sum(-1)
```

---

**Algorithm 2:** Deterministic GMM CDF indexing (CDFIndex)

**Input:** symbol $\hat{y}$, CDF index $i_1 \ldots i_K$, round-down means $\lfloor \mu_1 \rfloor \ldots \lfloor \mu_K \rfloor$, quantized weights
$\qquad q_{\pi_1} \ldots q_{\pi_K}$
**Output:** CDF value $c_{\hat{y}}$
**Data:** CDF LUTs $\mathrm{CDF}[0], \ldots, \mathrm{CDF}[64]$, upper bound of frequency cumulate $\mathrm{CDF}_{\max}$, range
$\qquad$ bound $R$

1  $c_{\hat{y}} = 0$
2  **for** $k$ *in* $1 \ldots K$ **do**
3  $\quad$ $p = \hat{y} - \lfloor \mu_k \rfloor$
4  $\quad$ $C_k = \mathrm{CDF}[i_k]$
5  $\quad$ **if** $p \geq R$ **then**
6  $\quad\quad$ $c_k := \mathrm{CDF}_{\max}$
7  $\quad$ **else if** $p \leq -R$ **then**
8  $\quad\quad$ $c_k := 0$
9  $\quad$ **else**
10 $\quad\quad$ $c_k := C_k[p + R]$ $\quad$ `// shift R to obtain non-negative query index`
11 $\quad$ $c_k := q_{\pi_k} \times c_k$
12 $\quad$ $c_{\hat{y}} := c_{\hat{y}} + c_k$

---

**Algorithm 3:** Encoding with deterministic GMM

**Input:** symbol $\hat{y}$, presently coder state $t$, CDF index $i_1 \ldots i_K$, round-down means
$\qquad \lfloor \mu_1 \rfloor \ldots \lfloor \mu_K \rfloor$, quantized weights $q_{\pi_1} \ldots q_{\pi_K}$
**Output:** updated state $t$
**Data:** upper bound of frequency cumulate $\mathrm{CDF}_{\max}$

1  $c_{\hat{y}} := \mathrm{CDFIndex}(\hat{y}, i_1 \ldots i_K, \lfloor \mu_1 \rfloor \ldots \lfloor \mu_K \rfloor, q_{\pi_1} \ldots q_{\pi_K})$
2  **if** $c_{\hat{y}} = \sum_{k=1}^{K} q_\pi \mathrm{CDF}_{\max}$ **or** $c_{\hat{y}} = 0$ **then**
3  $\quad$ $c_{(1-R)} := \mathrm{CDFIndex}(-R + 1, i_1 \ldots i_K, \lfloor \mu_1 \rfloor \ldots \lfloor \mu_K \rfloor, q_{\pi_1} \ldots q_{\pi_K})$
4  $\quad$ $\mathrm{GolombEnc}(\hat{y}, t)$
5  $\quad$ $t := \mathrm{ArithmeticEnc}(-R, 0, c_{(1-R)}, t)$
6  **else**
7  $\quad$ $c_{\hat{y}+1} := \mathrm{CDFIndex}(\hat{y} + 1, i_1 \ldots i_K, \lfloor \mu_1 \rfloor \ldots \lfloor \mu_K \rfloor, q_{\pi_1} \ldots q_{\pi_K})$
8  $\quad$ $t := \mathrm{ArithmeticEnc}(\hat{y}, c_{\hat{y}}, c_{\hat{y}+1}, t)$

---

---

**Algorithm 4:** Decoding with deterministic GMM

---

**Input:** presently decoder state $t$, CDF index $i_1 \ldots i_K$, round-down means $\lfloor \mu_1 \rfloor \ldots \lfloor \mu_K \rfloor$,
quantized weights $q_{\pi_1} \ldots q_{\pi_K}$
**Output:** decoded symbol $\hat{y}$, updated state $t$
**Data:** upper bound of frequency cumulate $\text{CDF}_{\max}$

```
/* Search-based ŷ decoding.  This can be replaced by a binary
   search                                                      */
```
1   $c_{\text{lo}} := \text{GetDecoderCDFLow}(t)$
2   $\hat{y} := -R$
3   **for** $y$ *in* $-R \ldots R$ **do**
4     $c_y := \text{CDFIndex}(y, i_1 \ldots i_K, \lfloor \mu_1 \rfloor \ldots \lfloor \mu_K \rfloor, q_{\pi_1} \ldots q_{\pi_K})$
5     $c_{y+1} := \text{CDFIndex}(y + 1, i_1 \ldots i_K, \lfloor \mu_1 \rfloor \ldots \lfloor \mu_K \rfloor, q_{\pi_1} \ldots q_{\pi_K})$
6     **if** $c_y \leq c_{\text{lo}} < c_{y+1}$ **then**
7       $\hat{y} := y$

```
/* Update the decoder state                                   */
```
8   **if** $\hat{y} = -R$ **then**
```
    /* The Golomb coding is adopted                           */
```
9     $c_{(1-R)} := \text{CDFIndex}(-R + 1, i_1 \ldots i_K, \lfloor \mu_1 \rfloor \ldots \lfloor \mu_K \rfloor, q_{\pi_1} \ldots q_{\pi_K})$
10    $t := \text{ArithmeticDecUpd}(0, c_{(1-R)}, t)$
11    $\hat{y} := \text{GolombDec}(t)$

12 **else**
13    $c_{\hat{y}} := \text{CDFIndex}(\hat{y}, i_1 \ldots i_K, \lfloor \mu_1 \rfloor \ldots \lfloor \mu_K \rfloor, q_{\pi_1} \ldots q_{\pi_K})$
14    $c_{\hat{y}+1} := \text{CDFIndex}(\hat{y} + 1, i_1 \ldots i_K, \lfloor \mu_1 \rfloor \ldots \lfloor \mu_K \rfloor, q_{\pi_1} \ldots q_{\pi_K})$
15    $t := \text{ArithmeticDecUpd}(c_{\hat{y}}, c_{\hat{y}+1}, t)$

---

