# OpenReview forum: "Post-Training Quantization Is All You Need to Perform Cross-Platform Learned Image Compression"
_ICLR.cc/2022/Conference — ICLR 2022 Submitted_

### Official Review · Reviewer_GrpS · 2021-10-30

**Correctness:** 4
**Technical Novelty And Significance:** 2
**Empirical Novelty And Significance:** 3
**Recommendation:** 6
**Confidence:** 5

**Main Review:**

Strengths:
The authors do a great job of immediately conveying the problems in image compression (designing deterministic decoders and the speed performance of those decoders).

The results are strong, they show very little to no loss until high bitrates due to this quantization technique.

The reforming of the entropy parameter discretization to be more computationally efficient shows impressive runtime improvements on this quantization process with no RD loss.

Additionally, the appendices were well written and contain many additional derivations and detailed descriptions of the quantization algorithms applied.


Weaknesses:
Rate-distortion performance is rarely shown outside the 1.0 bpp range. Since the losses appear to be more evident at higher bitrate, it would be instructive to know if PTQ is as successful outside of the low to medium quality range.

How much of a difference does one LUT make? The baselines are compared against a 64 level table and the new method proposes a much faster 65 level table. It is an obvious improvement in runtime due the formulation, but are there any significant RD gains for having another level.


Questions/Feedback:
Table 2: I'm not 100% sure what is being compared here. Is this a hyper synthesis with scale, scale-mean, context, GMM? Is there a reference for the popularly vectorized version? Was your implementation done in C++ or inline Assembly to guarantee the fast assembly instructions were properly generated?

In Section 3.1: "a grid search minimizing the reconstruction error", how much does this grid search help in the quantization process? Does this need to be done on a per loss function basis or per model? i.e. if I train a model for a specific lambda with a specific architecture, will I generally use the same quantization step, or is this more sensitive to individual runs in compression models?

**Summary Of The Paper:**

Quantization issues around entropy coding in neural image compression can lead to non-determinism across platforms and runtimes (i.e. CPU to GPU or even different drivers on the same GPU). Many works in the are perform quantization during training, which can make for more complex or longer training setups. This paper shows the effectiveness of post-training quantization techniques, shows the results are successful even on context and GMM models, and introduces a faster entropy parameter discretization method.

**Summary Of The Review:**

This is a well written paper that shows the effectiveness of existing post training quantization techniques when applied to multiple neural image architectures. In addition to showing how well this PTQ techniques perform, a more computationally efficient entropy parameter discretization technique is introduced.

---

> ### Author Response · Authors · 2021-11-21
> **Response to Reviewer GrpS**
>
> ## Results outside 1.0 BPP
>
> Thanks for your suggestion. We now add the performance for bpp > 1 in Figure 5b. Our method performs the best. The difference between ours and fp32 model is still negligible (please also refer to Table 1 for quantitative BD-rate results).
>
>
> ## One additional level of LUT
>
> Introducing one more LUT level is for the convenience of the derivation in section 4.1. It marginally influences the RD performance as the 65-th level represents the largest sigma value which stands for an extremely rare situation. As mentioned in the paragraph after eq.11, the 64 level baseline (sigma_min = 0.11, sigma_max = 256) is very different in value ranges from our 65 level method (sigma_min = 0.125, sigma_max = 32). Larger sigma values are rarely used and the representation accuracy of the largest sigma has negligible influence (e.g. our sigma_max is 32 and is much smaller than the original 256, however, no performance deterioration is observed). Note that the key designing principle of sigma discretization (either into 64 or 65 levels) is to retain the coding performance of using continuous sigma values, having one extra level will not cause any unfair comparison since we even use different sigma value ranges.
>
>
>
> ## Table 2 (Table 3 in the updated revision) explanation
>
> The *hyper synthesis* just denotes the average running time of the hyper synthesizer (please see Figure 2, it’s a single network not related to scale, scale-mean, context, or GMM) on GTX 1060. We provide this running time of hyer synthesizer to show the importance of improving STD discretization speed. If we adopt a comparison-based discretization, it can be as slow as the hyper synthesizer inference and become a speed bottleneck. All the implementations in this Table are based on python for a fair comparison. Further optimization of each discretization method using c++ is out of the bound of this paper. However, we do try to optimize the comparison-based approach in CompressAI and previous works by providing the results of a vectorized version. Now we add the vectorized version in Appendix E.3 for readers’ reference.
>
> ## Grid search
>
> The grid search is a part of the general PTQ technique. Please see reference (Nagel et al., 2021). We just frankly adopt it in our investigation. It is per-layer adopted after the model training and it is costless. In our experiments, it takes about two minutes to search the quantization steps for all layers of Minnen2018. For your question, yes, generally we need to conduct grid search per model architecture and per loss function. The same architecture trained with different loss functions will produce models with very different weights, thus the quantization step cannot be shared.

---

### Official Review · Reviewer_oV3R · 2021-11-02

**Correctness:** 2
**Technical Novelty And Significance:** 2
**Empirical Novelty And Significance:** 3
**Recommendation:** 3
**Confidence:** 4

**Main Review:**

It is interesting that post-training quantization is enough for the model robustness of learned image compression.

However, this paper requires polishing for the presentation and comparisons and the writing clarity should be improved.
For example, the term 'cross-platform inconsistency' is hard to understand until reading the Appendix and the paper of Balle et al. (2019).
Figure 1 and 2 present the problem statement and the methods in previous works which limit to emphasize the contributions of this paper.
Experiments do not analyze the contributions of the proposed method described in Section 3.
More importantly, this paper misses the comparisons to the method of Balle et al. (2019) for the model robustness across devices.

On the other hand, the proposed method described in Section 4 seems to be an alternative to post-training quantization since it is a predetermined CDF for different devices that minimizes the cross-platform inconsistency.
Based on the alternative, I'm curious about the importance of the cross-platform inconsistency which can easily be solved by saving the same CDF as metadata and using it across devices.

While quantizing weight with per-channel scaling factors, which granularity is used for the feature map quantization?
Does the proposed quantized model have actual latency acceleration with the per-channel weight quantization?



**Summary Of The Paper:**

This paper presents a network quantization method for robust learned image compression.
Recent deep-learning-based approaches for image compression usually adopt the cumulative distribution functions (CDF) for both encoding and decoding processes.
However, the CDFs are usually implemented in devices (e.g., CPU, GPU) with different approximations.
The different values from approximated CDFs incur significant decoding errors.
This paper alleviates this problem by quantizing the feature maps and weights of the encoder and decoder to 8-bit.

**Summary Of The Review:**

This paper presents a simple practical solution, post-training quantization, for robust learned image compression using neural networks.
However, the contributions of this paper are marginal since the problem statement and solution are overlapped with the previous work (Balle et al., 2019).
Moreover, this paper presents limited comparisons to the work of Balle et al. (2019) while Section 1 describes its failure cases.
The authors should thoroughly state the differences and improvements of the proposed method to the work of Balle et al. (2019).

---

> ### Author Response · Authors · 2021-11-21
> **Response to Reviewer oV3R**
>
> ## Writing clarity and Figure 1, 2
>
> Yes, Fig. 1 and Fig. 2 present the problem statement and model architecture proposed in previous works. We think Fig. 1 is important for understanding the cross-platform inconsistency issue, and without Fig. 2 it will be very difficult to describe the network architecture, context model, and all the related notations. So we think Fig. 1 and Fig. 2 are necessary for the readers to understand our work. We have tried to make it easier to understand for readers from both learned image compression and model quantization. We provide the necessary information in the introduction and the preliminary part, where we explain in detail why we need cross-platform consistency. As other reviewers agree, our writing is self-contained. We agree Balle et al. (2019) is helpful to understand our work as it is the first paper addressing this cross-platform inconsistency issue.
>
> ## Analyze the method in sec.3
>
> The method described in sec.3.1 is from the standard  PTQ approach.  As these techniques are viewed as baseline methods in the PTQ community (Nagel et al., 2021), we think we do not need to provide additional results for the ablation study. In Appendix C.4, we have compared the performance of using (or not) different weight rounding reconstruction approaches. Sec. 3.2 provides an adaptation to the requantization method proposed for QAT (Jacob et al. (2018)). We slightly change the calculation of n and m_0 to avoid the use of a 64-bit multiplication result register (which normally requires special hardware support) in Jacob et al.(2018). Note that our method of calculating n and m_0 results in larger representation error compared with Jacob et al. (2018), however, our experiments in Fig. 5 show that our approach of calculating m_0 and using 32-bit accumulation is sufficient for retaining the RD performance of determinized models. We think no extra results need to be provided to support this contribution.
>
> ## Comparison with Balle et al. (2019)
>
> Note that Balle et al. (2019) is a dedicated QAT method for solving the cross-platform inconsistency issue in learned compression. As emphasized in the introduction part, we have at least two major contributions to this problem: 1) we provide a convenient PTQ solution and prove its effectiveness in determinizing context-model-involved architectures like Minnen et al. (2018), which has not been considered before. 2) we propose a novel approach to discretize the entropy parameters. With these two major contributions, we believe our work is obviously different from Balle et al. (2019). We agree that we should provide the failure case of Balle et al. (2019) to demonstrate our improvement in RD performance. We have successfully reproduced the good performance reported in Balle et al. (2019) for determinizing Balle et al. (2018). However, although we try hard, we find this method cannot keep marginal performance deterioration for determinizing context-model-involved architectures like Minnen et al. (2018). Actually, when applied to Minnen et al. (2018), the training method in Balle et al. (2019) is very unstable. We now add this result in Fig. 5b, where our method performs the best. Moreover, our solution is more compatible with popular hardware because we adopt a more standard quantization scheme.
>
> ## Parameter discretization seems like an alternative to PTQ.
>
> The two techniques proposed in sec. 3 and 4 address two different issues. PTQ ensures the deterministic inference of entropy estimation models, and parameter discretization generating LUT is for deterministic arithmetic encoding/decoding after the model inference. Please refer to Appendix D.2 and D.3, where we thoroughly explained why this discretization is necessary.
>
> ## The importance of inconsistency issue
>
> Following the previous comment, we cannot easily solve this inconsistency by saving CDF as metadata in the LUT. LUT is used to store pre-computed CDFs while deterministic model inference is used to ensure that the index querying the LUT (during encoding and decoding) is calculated deterministically. Note that the querying index cannot be pre-computed.  Without the deterministic model inference (i.e. using integer network), the floating round-off error is almost inevitable, which will cause the query to wrong LUT (as explained in Fig. 13) and the decoding will fail like Fig. 1(b).
> ## Quantization granularity
>
> Please see sec.3.1. We use per-tensor activation quantization, for the feature map. We adopt per-channel quantization on weights only. This setting is well supported by existing hardware, e.g. TensorRT or DSP. It is a popular setting adopted by quantization literature (Nagel et al., 2021). Though **our major contribution is not about inference acceleration**, this well-supported setting can indeed help speed up the inference, which is yet out of the bound of this paper (it is well-known that platforms like TensorRT can speed up inference by using per-channel quantized weight).

---

> > ### Comment · Reviewer_oV3R · 2021-11-26
> > **Response to the authors**
> >
> > After carefully reading the rebuttal, my concerns about this paper, limited contributions to the previous works, are not alleviated.
> >
> > Specifically, using PTQ techniques for learned image compression methods can be regarded as having the following sub-contributions,
> >
> > 1) Quantized weight and activation of learned compression methods can reduce decoding error on cross-platform.
> > However, this contribution is already presented in Balle et al. (2019).
> >
> > 2) PTQ is a general technique than Balle et al. (2019).
> > However, the experiments only report the comparisons of a single model from Minnen et al. (2018).
> >
> > 3) PTQ is easy to apply than the training techniques in Balle et al. (2019).
> > However, the per-channel quantization of the proposed method incurs computational overhead [A] while the networks in Balle et al. (2019) enable low-bit operations using integer values.
> > Note that the experiment does not report thorough latency or computation cost comparisons of the end-to-end compression process to Balle et al. (2019), while Table 3 presents the latency of discretization.
> >
> >
> > Also, the experimental results do not report the effectiveness of the proposed discretization method for the entropy parameters in terms of overall decoding error for cross-platform image compression.
> >
> > I keep my score.
> >
> > [A] Vs-quant: Per-vector scaled quantization for accurate low-precision neural net-work inference. In MLSys, 2021

---

> > > ### Author Response · Authors · 2021-11-26
> > > **Appendant Response to Reviewer oV3R, Part 1**
> > >
> > > Thanks for the advice. Here we further explain our technical contributions to address the concerns.
> > >
> > > **Contribution is already presented in Balle et al. (2019).** Please note that we did not claim ‘Quantized weight and activation of learned compression methods can reduce decoding error on cross-platform’ as our contribution. As explained in our manuscript (the introduction part) and agreed by other reviewers, our work is obviously different from Balle et al. (2019).  We believe we have at least two major contributions: 1) we provide a convenient PTQ solution and prove its effectiveness in determinizing context-model-involved learned image compression (LIC) architectures like Minnen et al. (2018) and Cheng et al. (2020), which has not been considered before. 2) we propose a novel approach to discretize the entropy parameters, which also runs faster (Table 3). The modification to existing integer-only requantization in section 3.2 and the GMM indexing method in section 4.2 should also be considered as non-trivial technical contributions, which is not considered in previous works. Moreover, our solution is much more compatible with popular hardware because of a simpler and more general quantization scheme.
> > >
> > > In conclusion, we believe we have sufficient novelty and contribution in this work, regrading both problem setting (determinizing context-model-involved LIC) and technical contribution (hardware friendly PTQ for consistent LIC, hardware friendly 32-bit integer-only requantization, novel parameter discretization, and GMM indexing)
> > >
> > > ---
> > >
> > > **Only report the comparisons to Balle et al.(2019) on Minnen et al. (2018).** We evaluate Balle et al. (2019) on Minnen et al. (2018) and we think it is enough to explain the superiority of our methods, which is also agreed by other reviewers. Minnen et al. (2018) propose a representative architecture that includes the mean-scale Gaussian entropy model and context modeling technique. Note that the Balle et al. (2019) approach is designed only for the earlier proposed models without context modeling. It is not designed for context-involved architectures.  Minnen et al. (2018) is a single model but represents a very general context-modeling architecture with SOTA performance which is still adopted in the newest works (Guo et al., 2021; Chen et al., 2021; Xie et al., 2021; Gao et al, 2021).
> > >
> > > **Ref**
> > >
> > > Causal contextual prediction for learned image compression, Guo et al, IEEE  Transactions on Circuits and Systems for Video Technology, 2021.
> > >
> > > End-to-end learnt image compression via non-local attention optimization and improved context modeling, Chen et al., TIP, 2021.
> > >
> > > Enhanced invertible encoding for learned image compression, Xie et al., ACMMM 2021.
> > >
> > > Neural image compression via attentional multi-scale back projection and frequency decomposition, Gao et al., ICCV, 2021.

---

> > > ### Author Response · Authors · 2021-11-26
> > > **Appendant Response to Reviewer oV3R, Part 2**
> > >
> > > **Overhead of per channel weight quantization.** We apply per channel quantization only for the weight, which is very popular in the model quantization community (Nagel et al., 2021) and is well supported by popular inference engines including TensorRT and SNPE. Note that the scaling vector c in eq.2 of Balle et al. (2019) is not a scalar, which means **the quantization method of Balle et al. (2019) is per-channel in nature instead of per-tensor**.
> > >
> > > ---
> > >
> > > **Computation cost comparisons with Balle et al. (2019)** Please note that we use the most popular uniform affine quantization along with per-channel weight quantization，which is readily supported by popular inference engines including TensorRT (for GPU) and SNPE (for DSP). **As any other model quantization literature, our network inference is conducted using low-bit integer operations, which is not the unique advantage of Balle et al. (2019).** Actually, the quantization scheme in Balle et al. (2019) is dedicated and not readily supported by current hardware and inference engine due to uncommonly used bit-widths (i.e. the hybrid use of int8, uint8, int32, and uint32).
> > >
> > > Considering the above arguments, compared with Balle et al. (2019), our method does not introduce any overhead to current model quantization techniques and low-bit inference hardware. It is obvious our method is easier to be deployed regarding popular inference engines and hardware and can leverage the benefit of low-bit integer operations.
> > >
> > >  ---
> > >
> > > **Effectiveness of entropy parameter discretization in terms of decoding error.** The entropy parameter discretization is used to generate LUT. LUT is used to store pre-computed CDFs deterministically (Appendix D.2 and D.3) while model quantization (PTQ) is used to ensure that the index querying the LUT (during encoding and decoding) can be calculated deterministically. If we only use model quantization without entropy parameter discretization, the CDFs we query may be cross-platform inconsistent (as explained in Appendix D.2). If we only use LUT to save the same CDFs across platforms, we may obtain a non-deterministic querying index from the floating-point networks and we will query the wrong LUT (as explained in Figure 13). **In both cases, the decoding might fail**. As reported in Table 2, we have zero cross-platform decoding error, proving the effectiveness of both model quantization and entropy parameter discretization for generating LUT.  Note that for cross-platform decoding, the error rate makes no sense as we must ensure zero decoding error (which requires model quantization, entropy parameter discretization, and LUT being used together), otherwise we may lose the data permanently. **The only target is to achieve zero decoding error, which we have reported in Table 2**.

---

> > > ### Comment · Reviewer_oV3R · 2021-11-26
> > > **Response to the authors**
> > >
> > > I understand your explanation about the proposed method, but your claims have limited evidence supported by experiments.
> > >
> > >
> > > 1) It is not obvious that the good performance on the model from Minnen et al. (2018) guarantees good performances on the other context-modeling-based models. More experiments should support this claim.
> > >
> > > 2) It is not obvious that the proposed method has no computational overhead than Balle et al. (2019).
> > > Balle et al. (2019) used integer scaling factors, while Eq. (14) does not describe the bit-width of scaling factors (s_w).
> > > Floating-point scaling factors require additional quantization operation for features with floating-point scaling.
> > > If the proposed method uses integer scaling factors like Balle et al. (2019), the ablation study of the proposed techniques, which is missed in the manuscript, becomes more important to verify which technique improves the performance.
> > >
> > > 3) The two cases, you mentioned for entropy parameter discretization, are not obvious, where experimental results should support them.
> > >
> > >
> > > This paper requires thorough analyses and experiments since this is not the first work to quantize networks for cross-platform inconsistency.

---

> > > > ### Author Response · Authors · 2021-11-26
> > > > **Response to Reviewer oV3R**
> > > >
> > > > **Good performances on the other context-modeling-based models.** Note that we also present the good performance on Cheng et al. (2020), as shown in Fig.5(a). Furthermore, we present the performance of the two models using parallel checkerboard adaption (He et al., 2021) in the Appendix. Various recent approaches follow these two widely-used baselines and change only the main transform of them (e.g. Xie et al., 2021; Gao et al., 2021), which does not matter with our approach. We believe the RD curves of both Minnen et al. (2018) and Cheng et al. (2020) can already demonstrate the good performance on context-modeling-based models.
> > > >
> > > > ---
> > > >
> > > > **Scaling-factor s_w results in no overhead.** We highlight Figure 3, though s_w is floating-point, the integer-arithmetic-only requantization omits any floating-point operators and replaces them by integer multiplications and bit-shifts (for a baseline version of this idea, please refer to the classic model quantization paper Jacob et al., 2018). One of the most important points of our paper is to avoid using any floating-point arithmetic.
> > > >
> > > > For comment ‘If the proposed method uses integer scaling factors like Balle et al. (2019), the ablation study of the proposed techniques becomes more important to verify which technique improves the performance’.   If we understand correctly, the reviewer wants to know why our PTQ method performs better than the QAT approach in Balle et al. (2019).  In our experiment, we find the method in Balle et al. (2019) is very difficult to train when applied on context-involved LIC. We think the dedicated quantization scheme in Balle et al. (2019) should also have enough capacity or precision to achieve similar results as our method. However, the training is very unstable and the capacity cannot be utilized sufficiently.  Since our quantization scheme is already simple and standard, we think it is not necessary (and also out of the bound) to investigate whether PTQ can fully utilize the capacity/precision of the dedicated quantization scheme in Balle et al. (2019). Thanks for this question, we think it is meaningful to investigate QAT and PTQ again in the future, to obtain cross-platform consistency as well as fast running speed (especially for the main transform which occupies most of the running time), where the quantization scheme and bit-width can be reconsidered. Hope this can clarify the reviewer’s concern.
> > > >
> > > > ---
> > > >
> > > > **Two cases discussed for explaining entropy parameter discretization**. The previously published literature, Balle et al. (2019) and Sun et al. (2021), also view the parameter discretization as **an inevitable part** of the entire determinization process. Both of them did not present the ablation study about its effectiveness to the decoding error. We believe this accompanied usage of model quantization and parameter discretization is obvious to the learned image coding community.
> > > >
> > > >
> > > >
> > > > After all, the parameter discretization itself is not our invention, and we do not claim using entropy parameter discretization as our contribution. Our claim is:
> > > > > We propose a novel approach to discretize the entropy parameters, which can be computed directly in a deterministic manner. Compared with the existing method based on the searching algorithm, our method significantly speeds up the parameter discretization and eliminates the bottleneck.
> > > >
> > > > which is supported by experiments in Tables 2 and 3 (i.e. zero decoding error and faster parameter discretization speed).

---

### Official Review · Reviewer_L7dn · 2021-11-02

**Correctness:** 4
**Technical Novelty And Significance:** 2
**Empirical Novelty And Significance:** 3
**Recommendation:** 6
**Confidence:** 4

**Main Review:**

Strengths:
1.  It is interesting to see the model compression techniques could be beneficial for the learned image compression in terms of cross-platform development. Instead of designing new model components, the proposed approach is more flexible.
2. The experimental results are encouraging. It seems that this quantization method does not introduce too much performance degradation.

Weaknesses:
Since the major techniques are borrowed from the quantization techniques in model compression, so the technical contribution may be limited.  The authors are suggested to make a clear description for the difference between the techniques used in this paper and that in model compression.

Besides, the authors only provide the RD curve below ~1bpp. How about the compression performance at high bitrate?

**Summary Of The Paper:**

The paper tried to solve the non-deterministic issue for the learned image compression and reduce the inconsistent cross-platform probability prediction. It is a practical problem in the development of learned image compression. The authors utilize some off-the-shelf techniques in the model compression and revisit the non-deterministic issue from the perspective of model compression. The experimental results are convincing.

**Summary Of The Review:**

I tend to accept this paper though I have some concerns about the technical novelty.

---

> ### Author Response · Authors · 2021-11-21
> **Response to Reviewer L7dn**
>
> ## Clarification on the difference from model compression
>
> Thanks for the suggestion. We have emphasized in the ending of section 1 that we are not aiming to improve quantization techniques. We aim to simplify the study on the non-determinism issue in learned image compression (LIC) by exploiting quantization techniques, especially PTQ.  As for the quantization community, we have described a contribution to integer-only requantization in section 3.2, which can avoid the need for a 64-bit multiplication result register.  We think this point should be non-trivial.
>
> However, for the LIC community, we believe we have at least two major contributions: 1) we provide a convenient PTQ solution and prove its effectiveness in determinizing context-model-involved architectures like Minnen et al. (2018), which has not been considered before. 2) we propose a novel approach to discretize the entropy parameters, which also runs faster (Table 3). The modification to existing integer-only requantization in section 3.2 and the GMM indexing method in section 4.2 should also be considered as non-trivial technical contributions, which is not considered in previous works including Balle et al. (2019) and Sun et al. (2020, 2021). Moreover, our solution is much more compatible with popular hardware because of a simpler and more general quantization scheme (we do not need per-channel quantization for activation which is seldom used in the quantization community because of extra overhead ((Nagel et al., 2021)).
>
> In conclusion, we believe we have sufficient novelty in this work, regrading both problem setting (determinizing context-model-involved LIC) and technical contribution (hardware friendly PTQ for consistent LIC, hardware friendly 32-bit integer-only requantization, novel parameter discretization, and GMM indexing).
>
>
>
>
> ## RD curve >1 BPP
>
> We now add the performance for bpp > 1 in Figure 5b. Our method performs the best. The difference between ours and FP32 model is still negligible (please also refer to Table 1 for quantitative BD-rate results).

---

### Official Review · Reviewer_eyVf · 2021-11-02

**Correctness:** 2
**Technical Novelty And Significance:** 2
**Empirical Novelty And Significance:** 3
**Recommendation:** 6
**Confidence:** 3

**Main Review:**

Pros:

1. Authors experimentally showed that PTQ can be used to prevent data corruption (Table 1). The effect on RD performance is also shown in Figure 6.
2. I think cross-platform support for learned image compression is an important issue for practical applications.


Cons:

1. Issues to be addressed are unclear:
- The authors state in chapter 1: "However, we find that on more complex models with context modeling like Minnen et al. (2018) and Cheng et al. (2020), adopting this integer network approach cannot keep the performance loss negligible. ", it is interesting but I could not find evidence for this; if the RD rate performance of the integer network is degraded, it should be shown in experiments.

2. Comparisons with previous methods (e.g. Balle ́ et al. (2019) and Sun et al. (2021)) are not sufficient:
- If there are any advantages of this research compared to the previous methods (e.g., if previous methods are not able to prevent data corruption in some image compression models), I think you should clearly show that comparison.
- This paper proposes a new method (Binary Logarithm based STD discretization) to aim for more hardware-friendly. It is interesting but unfortunately, the experiments in Table 2 do not seem to show a comparison with Sun et al. (2021), which is written like a baseline in chapter 4.

3. Other unclear points:
- Is Table 1 the result of applying the LUT (proposed method in chapter 4)? If the error rate is 0% with only PTQ without applying the LUT, why is LUT necessary? Or, if you were applying the LUT, what happens if you do not apply the LUT but apply the PTQ? I am curious about the contribution for prevent data corruption of each of the PTQ and LUT.
- Is the time shown in Table 2 the average time for the entire data set?

**Summary Of The Paper:**

Learned image compression has a known issue that the nondeterminism of the floating point operation makes it impossible to recover the compressed data on different platforms.

This paper experimentally shows in compression models using context modeling (Minnen et al. (2018)) that it is possible to recover the data on different platforms even when using the known quantization method called PTQ. (Table.1 )

Referring to the method of computing STD using pre-computed sampling points σˆ by Sun et al. (2021), the authors proposed a method called Binary Logarithm based STD discretization.

**Summary Of The Review:**

The authors experimentally showed that PTQ can be used to prevent data corruption of learned image compression, but the value of this research is not clear due to the unclear target issues, insufficient comparison with previous methods (e.g. Balle ́ et al. (2019) and Sun et al. (2021)), and some unclear points, so I considered that currently this paper did not meet the conditions for acceptance.

---

> ### Author Response · Authors · 2021-11-21
> **Response to Reviewer eyVf**
>
> ## Cons.1 & Cons.2.1
>
> Yes, we should have provided these results for better comparison. We now update Figure 5b. We have successfully reproduced the good performance reported in Balle et al. (2019) for determinizing Balle et al. (2018). However, although we try hard, we find this method cannot keep marginal performance deterioration for determinizing context-model-involved architectures like Minnen et al. (2018). Actually, when applied on Minnen et al. (2018), the training method in Balle et al. (2019) is very unstable. We also implement Sun et al. (2021) and add the results in Figure 5b. We can see that per-channel quantization of activations is required for Sun et al. (2021) to achieve better performance on Minnen et al. (2018).  Our proposed method performs the best and is more compatible with popular hardware because the activations are quantized in a per-tensor manner.
>
>
>
> ## Cons.2.2 Explanation to Table 2 and comparison to Sun et al. (2021)
>
> Please see the discussion at the beginning of section 4.1, just below eq.10, "Sun et al. (2021) obtains i_sigma ... by **comparing** the fix-point sigma value with precomputed sampling points". Which exactly corresponds to the **Comparison** columns in Table 3. This comparison-based approach can be localized to the implementation of tensorflow-compression[1] and CompressAI[2]. Since we do not find the official implementation of Sun et al.(2021), we instead test the CompressAI implementation, which is almost the same as early tensorflow-compression implementation. We even find this for-loop-based implementation is sub-optimal and provide a faster implementation (the vectorized column in Table 3), yet it is still slower than ours.
>
> ## Cons.3.1
>
> Roughly speaking, LUT is used to store pre-computed CDFs deterministically (Appendix D.2 and D.3) while PTQ is used to ensure that the index querying the LUT (during encoding and decoding) is calculated deterministically. Note that the querying index cannot be pre-computed. If we only use PTQ, then we will obtain the deterministic querying index, but the CDFs we query may be cross-platform inconsistent (as explained in Appendix D.2). If we only use LUT to save the same CDFs across platforms, we may obtain a non-deterministic querying index from the floating-point networks and we will query the wrong LUT (as explained in Figure 13). In both cases, the decoding might fail.
>
>
>
>
> ## Cons.3.2
>
> Yes, it is averaged over all 24 images in Kodak.
>
>  ---
>
> [1]  [tensorflow-compression implementation](https://github.com/tensorflow/compression/blob/v1.3/tensorflow_compression/python/layers/entropy_models.py#L869-L881)
>
> [2] [CompressAI implementation](https://github.com/InterDigitalInc/CompressAI/blob/v1.1.8/compressai/entropy_models/entropy_models.py\#L653-L658)

---

> > ### Comment · Reviewer_eyVf · 2021-11-25
> > **Response to the authors**
> >
> > Thank you for your thoughtful responses and updating the paper.
> > Your responses and updated paper have addressed some of my concerns and I raise my rating.

---

### Official Review · Reviewer_2XDr · 2021-11-03

**Correctness:** 3
**Technical Novelty And Significance:** 2
**Empirical Novelty And Significance:** 2
**Recommendation:** 6
**Confidence:** 3

**Main Review:**

## Strengths
   - In my view, the most important contribution of the paper is to demonstrate that state-of-the-art image compression methods can be appropriately quantized to retain essentially the same compression performance as their unquantized counterparts
   - The extension of the look-up table technique of [1] to Gaussian mixture models is interesting and clearly useful

## Weaknesses
   - I am not convinced that the authors can claim the application of post-training quantization (PTQ) to achieve cross-platform consistency for image compression models as a contribution. As I understand, this idea was first proposed in [2] and further developed in [1], and in both papers, the authors demonstrate the efficiency of PTQ. In fact, the authors seem to acknowledge this a couple of paragraphs above! I believe that what the authors can claim is the application of PTQ to models using autoregressive context models. Could the authors please clarify if my understanding is correct? In case it is, the claim should be changed to better reflect the contribution of the paper.
   - The results of the paper are not compared with relevant methods, most notably [1] and [2]. In particular, the authors present the performance of their method on the method of [3] but do not show the performance of e.g. [2] on the same model.
   - In the introduction, the authors make the following claim "However, we find that on more complex models with context modelling like Minnen et al. (2018) and Cheng et al. (2020), adopting this integer network approach cannot keep the performance loss negligible." - I could not find any reference to support this claim in the main text or the appendix. Could the authors point to any reference that would support this?

### Minor issues
   - The 1st and 2nd sentences of the abstract seem to be inconsistent with each other, the 1st implying that LIC is practical, while the 2nd implying that it is not.
   - There are many grammatical mistakes in the text that need to be fixed
   - The abbreviation "LUT" (which I believe stands for look-up table) is undefined in the text

## Questions
   - The authors claim that they use quantization not for the purpose of compression but as a way to switch calculations to integer arithmetic for cross-platform consistency. In this case, why do the authors still prefer 8-bit quantized representations for the activations and weights?
   - While the authors method to quantize the log-standard deviations is sensible, could they elaborate on why the method of [1] is not adequate? In the paper the authors claim that the method of [1] is moderately efficient, so I am unsure what benefit their method brings.

## After Rebuttal
The authors have addressed my main concerns, and I, therefore, decided to raise my score.


[1] H. Sun, L. Yu, and J. Katto. Learned image compression with fixed-point arithmetic. 2021 IEEE Picture Coding Symposium

[2] H. Sun, Z. Cheng, M. Takeuchi, and J. Katto, End-to-end learned image compression with fixed point weight quantization, 2020 IEEE International Conference on Image Processing

[3] J. Balle, D. Minnen, S. Singh, S. J. Hwang, and N. Johnston. Variational image compression with a scale hyperprior. ICLR 2018


**Summary Of The Paper:**

A well-known issue in learned compression is that due to inconsistencies in the implementations of low-level floating-point arithmetic across different hardware architectures, catastrophic decoding errors might occur for the compressed data. The authors propose several different ways to quantize the weights and activations of the models to make the encoding and decoding procedures use integer arithmetic only. Since the implementation of integer arithmetic is consistently implemented across different hardware, this solves the decoding issues, while also providing some speed-ups. The authors demonstrate the efficiency of their method by quantizing some state-of-the-art image compression methods and show that the performance of the quantized models does not degrade much.




**Summary Of The Review:**

I currently believe that while the work is useful to the community (see 1st point in "Strengths"), I think in its current form the paper makes a few considerable claims that are not supported.

If the authors can address my main concerns, I will reconsider raising my score.

---

> ### Author Response · Authors · 2021-11-21
> **Response to Reviewer 2XDr**
>
> ## Claim on using standard PTQ to deterministic LIC as the contribution
>
> Thanks for pointing out this. In [2] the activation is not quantized (as mentioned in its conclusion part and also stated in the introduction of [1]) and the inconsistency cannot be solved, so [2] cannot be viewed as prior work. As for [1], we actually have contacted the original authors by email and verified that channel-wise quantization is used for both weight and activation in the hyper synthesis network, which is rare in **general model quantization literature** because of extra overhead (Nagel et al., 2021). Additionally, the quantization scheme in [1] seems dedicated. Our quantization scheme based on uniform affine quantization is common in general model quantization and standard PTQ techniques.
>
> Also note that the publish date of [1] is 29 June - 2 July 2021, which is apparently after 5 June 2021, we can even consider [1] as contemporaneous according to ICLR 2022 guideline. Considering all the above arguments, we think our claim of ‘We evaluate and prove that, the deterministic computing issue of learned data compression can be reduced to **a general model quantization problem**. After applying **a standard post-training quantization (PTQ) technique**, we obtain cross-platform consistent image compression models with marginal compression performance loss.’ is proper.
>
>
>
>
> ## Comparison to prior works
>
> Thanks for your advice. Activations are not quantized in [2] and consistency cannot be guaranteed. We now add the results of Sun et al. (2021) in Figure 5b. We can see that per-channel quantization of activations is required for Sun et al. (2021) to achieve better performance on Minnen et al. (2018).  Our proposed method performs the best, where the activations are quantized in a per-tensor manner.
>
>
> ## Evidence Balle et al. (2019) not compatible with context models
>
>  We have successfully reproduced the good performance reported in Balle et al. (2019) for determinizing Balle et al. (2018). However, although we try hard, we find this method cannot keep marginal performance deterioration for determinizing context-model-involved architectures like Minnen et al. (2018). Actually, when applied on Minnen et al. (2018), the training method in Balle et al. (2019) is very unstable. We now add this result in Figure 5b.
>
>
> ## Why prefer 8bit
>
> Please see the beginning of section 3. 8-bit matrix multiplication and convolution is widely supported by various hardware platforms. So an 8-bit solution is more promising to be deployed on various hardware platforms.
>
>
>
> ## Claim on the efficiency of [1]
>
> The claim on the efficiency of Sun et al. (2021) is further demonstrated in Table 3. As stated in section 4.1, this claim says that Sun et al. (2021) obtains $\hat i_\sigma$ from parameter network output in a deterministic way by comparing the fix-point $\sigma$ value with pre-computed 64 sampling points $\hat \sigma$. We think this comparison process might be inefficient. As shown in Table 3, our direct calculation method is more efficient than comparison-based approaches. We have modified Table 3 caption to make this point clear.

---

> > ### Comment · Reviewer_2XDr · 2021-11-25
> > **Response to the authors**
> >
> > I thank the authors for their detailed rebuttal.
> >
> > The authors' rebuttals to the reviewers have addressed my main concerns with the work. In particular:
> > - The authors clarify how the prior art mentioned is different to their contribution and point out that the work of Sun et al. (2021) is contemporaneous to their work.
> > - The authors perform additional comparisons to prior work, in particular, they provide evidence for the claim made in their introduction that the method of Balle et al. (2019) does not work well with architectures that use context models.
> >
> > In light of the above, I decided to raise my score, as I now believe that this work would be a reasonable contribution to the literature.
> >
> > Some additional comments:
> > - I welcome the addition of Fig 5b, however, the authors should improve the caption provided for it, as it is very basic at the moment.
> > - The authors should define what BD rates are. I believe it is a method of comparing rate-distortion curves, but this should be explicitly stated in the text.

---

> > > ### Author Response · Authors · 2021-11-26
> > > **Appendant Response to Reviewer 2XDr**
> > >
> > > Thanks for the advice. The calculation of BD rate is described in section 5.1 as:
> > > > We present the RD results in Figure 5(b), and further report the corresponding BD-rates (Bjontegaard (2001)) in Table 1.
> > >
> > > We will improve the description in the final version.

---

### Author Response · Authors · 2021-11-26
**Paper1195 Author Response To Area Chairs And Reviewers**

We sincerely thank the area chairs and reviewers for their time and efforts.



Our paper focuses on a critical cross-platform inconsistency issue that prevents currently well-performed learned image compression models (LIC) from practical usage. Our experimental results show that the PTQ-based approach can well address the issue in a convenient way.



Our major contributions are 1) Provide a convenient PTQ solution and prove its effectiveness in determinizing context-model-involved LIC architectures like Minnen et al. (2018) and Cheng et al. (2020), which has not been considered before. 2) We propose a novel approach to discretize the entropy parameters, which also runs faster (Table 3). The modification to existing integer-only requantization in section 3.2 and the GMM indexing method in section 4.2 should also be considered as non-trivial technical contributions, which is not considered in previous works. Moreover, our solution is much more compatible with popular hardware because of a simpler and more general quantization scheme.



We believe we have sufficient novelty in this work, regrading both problem setting (determinizing context-model-involved LIC) and technical contribution (hardware friendly PTQ for consistent LIC, hardware friendly 32-bit integer-only requantization, novel parameter discretization, and GMM indexing).



We sincerely hope the rebuttal would convince our area chairs and reviewers to judge our paper fairly and render a convincing decision.

---

### Decision · Program_Chairs · 2022-01-20

**Decision:**

Reject

**Comment:**

The paper considers quantization issues for learned neural-network-based image compression methods.
Many works on the topic incorporate quantization into the training of the method. The paper provides evidence that post-training quantization is effective. Specifically, the paper demonstrates that state-of-the-art learned image compression methods can be quantized post-training and retain a very similar level of compression performance. The paper argues that this is important in particular for cross-platform applications, where an image is decompressed on different architectures. Finally, the paper proposes an approach to discretize entropy parameters.

The reviewers raised the following concerns.
- Reviewer 2XDr is concerned about the application of post-training quantization being a contribution, since post-training quantization has already been studied in [1] (and in the recent paper [2] that can be considered as concurrent work). The authors response is that the methods in [1] has extra overhead and clarify how the prior work is in fact different. This addressed the reviewer's concern, and the reviewer raised their score.

- Reviewer eyVf finds the comparisons with previous methods to be insufficient, and in general find the value of the research unclear, as the goals are not sufficiently specified. The authors clarified, and the reviewer was satisfied with the response and raised the rating to marginal above the threshold.

- Reviewer L7dn tends towards acceptance, but has concerns about the technical novelty, that are unspecified unfortunately.

- Reviewer oV3R argues that the solution is marginal relative to prior work, and votes to reject the paper. The authors responded why they think it isn't, and also wrote a private letter to the area chair in which they explain why they think that reviewer's oV3R should not be taken into account. I agree with the authors that the paper under review provides a step relative to Balle et al (2019), and that the writing of the paper is not an issue; however, the reviewer's overarching point is that the overall contribution is marginally significant when taking the prior work by Balle et al (2019) into account and this is the sentiment of other reviewers as well.

- Reviewer GrpS, an expert on image compression, leans towards acceptance and argues that the results are strong as they show little to no loss due to the quantization technique, but also rates the contribution to only be marginally significant and novel, and raises a few questions and issues, to which the reviewers responded.

This paper is really borderline. Four out of five reviewers rate this paper as marginally above the acceptance threshold. The consensus is that while the experiments and claims are correct, the contribution is only marginally significant or novel, in particular, relative to prior work, and therefore I recommend rejecting the paper. I would, however, not be upset if it would be accepted.